# The hominoid-specific gene *TBC1D3* promotes generation of basal neural progenitors and induces cortical folding in mice

Xiang-Chun Ju[1,2,3†], Qiong-Qiong Hou[1,2,3†], Ai-Li Sheng[1,2], Kong-Yan Wu[1,2], Yang Zhou[4], Ying Jin[4], Tieqiao Wen[5], Zhengang Yang[6], Xiaoqun Wang[7,8], Zhen-Ge Luo[1,2,3,7,9]*

[1]Institute of Neuroscience, Shanghai Institutes for Biological Sciences, Chinese Academy of Sciences, Shanghai, China; [2]State Key Laboratory of Neuroscience, Chinese Academy of Sciences, Shanghai, China; [3]Chinese Academy of Sciences University, Beijing, China; [4]The Institute of Health Sciences, Shanghai Institutes for Biological Sciences, Chinese Academy of Sciences, Shanghai, China; [5]School of Life Sciences, Shanghai University, Shanghai, China; [6]Institutes of Brain Science, State Key Laboratory of Medical Neurobiology, Fudan University, Shanghai, China; [7]CAS Center for Excellence in Brain Science and Intelligence Technology, Shanghai, China; [8]Institute of Biophysics, Chinese Academy of Sciences, Beijing, China; [9]ShanghaiTech University, Shanghai, China

*For correspondence: zgluo@ion.ac.cn

†These authors contributed equally to this work

Competing interests: The authors declare that no competing interests exist.

**Abstract** Cortical expansion and folding are often linked to the evolution of higher intelligence, but molecular and cellular mechanisms underlying cortical folding remain poorly understood. The hominoid-specific gene *TBC1D3* undergoes segmental duplications during hominoid evolution, but its role in brain development has not been explored. Here, we found that expression of *TBC1D3* in ventricular cortical progenitors of mice via *in utero* electroporation caused delamination of ventricular radial glia cells (vRGs) and promoted generation of self-renewing basal progenitors with typical morphology of outer radial glia (oRG), which are most abundant in primates. Furthermore, down-regulation of *TBC1D3* in cultured human brain slices decreased generation of oRGs. Interestingly, localized oRG proliferation resulting from either *in utero* electroporation or transgenic expression of TBC1D3, was often found to underlie cortical regions exhibiting folding. Thus, we have identified a hominoid gene that is required for oRG generation in regulating the cortical expansion and folding.

## Introduction

It is generally assumed that the expansion of the mammalian neocortex during evolution correlates with the increase in intelligence, and this process involves increased production of cortical neurons, resulting from an extended neurogenic period as well as increased proliferative ability of neural stem cells and progenitors (*Geschwind and Rakic, 2013*; *Lui et al., 2011*; *Sun and Hevner, 2014*; *Zilles et al., 2013*). To fit into a limited cranium, expanded cortical surfaces are folded to form gyri and sulci. Recent cross-species studies have shown the emergence of an outer subventricular zone (OSVZ) in the primate cortex, consisting of a massive pool of proliferating basal progenitors (BPs) and post-mitotic neurons (*Betizeau et al., 2013*; *Fietz et al., 2010*; *Hansen et al., 2010*;

**eLife digest** The outer layer of the mammalian brain the cerebral cortex plays a key role in memory, attention, awareness and thought. While rodents have a smooth cortical surface, the cortex of larger mammals such as primates is organized into folds and furrows. These folds increase the amount of cortex that can fit inside the confines of the skull, and are thought to have allowed the evolution of more advanced thought processes.

Mutations in various genes are likely to have contributed to the expansion and folding of the cortex. These mutations may not always have involved changes in the instructions encoded within the genes, but might instead have involved changes in the number of copies of a gene. One plausible candidate gene is *TBC1D3*, which is only found in the great apes and is active in the cortex. The chimpanzee genome contains a single copy of *TBC1D3* whereas the human genome contains multiple copies.

Ju, Hou et al. have now shown that introducing the *TBC1D3* gene into mouse embryos triggers changes in the embryonic cortex. Specifically, this gene increases the number of a type of cell called the outer radial glial cell in the cortex. These cells give rise to new neurons, and are usually rare in mice but abundant in the brains of animals with a folded cortex. Additional experiments using samples of human brain tissue confirmed that *TBC1D3* is required for the outer radial glial cells to form. The samples were collected from miscarried fetuses with the informed consent of the patients and following approved protocols and ethical guidelines.

Finally, introducing the *TBC1D3* gene into the mouse genome also gave rise to animals with a folded cortex, rather than their usual smooth brain surface. Further work is now required to identify how *TBC1D3* helps to generate outer radial glial cells, and to work out how these cells cause the cortex to expand. Testing the behavior of mice with the *TBC1D3* gene could also uncover the links between cortical folding and thought processes.

*Reillo et al., 2011*; *Smart et al., 2002*). Unlike the neuroepithelia-derived ventricular radial glial cells, which undergo repeated and typically asymmetric cell division at the apical surface of the ventricular zone, the BPs, after delamination from the apical surface, translocate to the SVZ, where they exhibit symmetric or asymmetric divisions. In primates, the recently identified outer (basal) radial glia (referred to as oRG or bRG) and the intermediate progenitors (IPs) in the OSVZ, which can undergo multiple rounds of symmetric or asymmetric divisions (*Betizeau et al., 2013*; *Hansen et al., 2010*), are two major forms of BPs. By contrast, the IPs and minimal oRG cells in the mouse SVZ usually exhibit final division to generate a pair of post-mitotic neurons (*Shitamukai et al., 2011*; *Wang et al., 2011*). The radial and lateral expansion of BPs is thought to be a main cause of cortical folding of gyrencephalic species (*Fietz and Huttner, 2011*; *Fietz et al., 2010*; *Hansen et al., 2010*; *Lewitus et al., 2014*; *Lui et al., 2011*; *Reillo et al., 2011*). In support of this hypothesis, forced expansion of BPs by down-regulating the DNA-associated protein Trnp1 or overexpressing cell cycle regulatory proteins Cdk4/Cyclin D1 resulted in gyrification of the cerebral cortex in naturally lissencephalic mouse or gyrencephalic ferret (*Nonaka-Kinoshita et al., 2013*; *Stahl et al., 2013*).

Given that genetic differences between humans and other species are likely to be the causes of human-specific traits, including complexity of cortical morphology, extensive studies have been performed in comparing genes and genetic elements of different species of primates and mammals (*Arcila et al., 2014*; *Fietz et al., 2012*; *Florio et al., 2015*; *Johnson et al., 2009, 2015*; *Kang et al., 2011*; *Konopka et al., 2012*; *Lui et al., 2014*; *Miller et al., 2014*; *O'Bleness et al., 2012*). In particular, several recent studies have aimed to uncover the distinctive transcriptional signature of the expanded human OSVZ or BPs that reside there, leading to the identification of a group of genes highly expressed in the human OSVZ (*Miller et al., 2014*), and human-specific orthologs preferentially expressed in human RGs (*Florio et al., 2015*; *Lui et al., 2014*; *Miller et al., 2014*; *Pollen et al., 2015*; *Thomsen et al., 2016*). For examples, platelet-derived growth factor D is expressed specifically and functionally important in human but not mouse RGs (*Lui et al., 2014*). A human lineage-specific Rho GTPase-activating protein could enhance the generation of IPs and cause neocortex expansion when expressed in the mouse brain (*Florio et al., 2015*). Since cortical folding emerges

progressively during primate evolution, multiple primate- and hominid-specific genes are likely to be involved in the emergence of cortical folding.

Gene duplication may play critical roles in brain evolution (*Geschwind and Rakic, 2013*). In particular, duplication of specific genes in humans may be responsible for the marked increase in cortical folding. The *TBC1D3* gene is derived from a segmental duplication, with multiple copies present in the human chromosome 17 and present in the chimpanzee genome as a single copy gene (but absent in other species) (*Hodzic et al., 2006*; *Pei et al., 2002*; *Perry et al., 2008*; *Zody et al., 2006*). Indeed, *TBC1D3* corresponds to one of the core duplicons that have been implicated in the expansion of intrachromosomal segmental duplications during hominoid evolution (*Jiang et al., 2007*). Because the timing of origination and amplification of the *TBC1D3* gene is consistent with the evolutionary divergence of primates (*Perry et al., 2008*; *Stahl and Wainszelbaum, 2009*), we decided to explore its role in brain development by expressing this gene in mice.

We found *TBC1D3* expression markedly elevated the generation and proliferation of BPs and resulted in extensive cortex folding in the mouse brain, and further delineated the molecular and cellular mechanisms underlying its action. Furthermore, *TBC1D3* is essential for the generation of BPs in cultured developing human brain slices. The transgenic mice generated in this study may provide a feasible model to link cortical folding to higher brain functions.

## Results

### TBC1D3 expression in mice delaminates ventricular neuroprogenitors

A previous study showed that TBC1D3 paralogues are expressed in most human tissues, including the brain (*Hodzic et al., 2006*). By using reverse transcription PCR, we found that the expression of TBC1D3 in the fetal human brain (gestational week, GW 26 to 40) was higher than that in the adult (*Figure 1A*). Immunofluorescence staining of cortical sections obtained from GW18 human specimens during the peak period of neurogenesis revealed high TBC1D3 expression near the ventricular surface and in the subventricular zone (SVZ) (*Figure 1B*), suggesting a role of TBC1D3 in cortex development.

To investigate the potential role of TBC1D3 in neural development, we introduced human TBC1D3 expression construct pE/nestin-TBC1D3 together with pCAG-YFP into neural precursors in the ventricular zone (VZ) of fetal mice at embryonic day 13.5 (E13.5), using *in utero* electroporation. At E15.5, in situ hybridization with an antisense *TBC1D3* probe showed that the transcript was expressed in TBC1D3-electroporated mice but not in control mice injected with the vehicle construct (*Figure 1C*), consistent with the absence of TBC1D3 in the murine genome (*Hodzic et al., 2006*). Interestingly, expression of TBC1D3 caused delamination of ventricular radial glia (vRG), and a decreased level of N-cadherin at adherens junctions (AJs) (*Figure 1D*) at the ventricular surface. The reduction in N-cadherin was probably due to either the transcriptional inhibition or destabilization of transcripts, because in situ hybridization and real-time PCR showed marked reduction of *N-cadherin (Cdh2)* transcript in VZ cells (*Figure 1—figure supplement 1A*) and flow cytometry-sorted YFP$^+$ cells expressing TBC1D3 (*Figure 1—figure supplement 1B–D*), respectively. Further analyses showed that expression of *TBC1D3* indeed caused destabilization of *Cdh2* mRNA in ReNeuron human neural progenitor cell line (*Figure 1—figure supplement 1E,F*). Since N-cadherin is known to be important for maintaining the alignment of radial glial cells at VZ (*Kadowaki et al., 2007*), reduction of N-cadherin due to TBC1D3 expression may be causally related to the delamination of ventricular neuroprogenitors. Indeed, we found that expression of *TBC1D3* caused dislocation of Numb and integrin beta 1 (ITGB1) which were originally polarized distributed in endfeet of vRGs (*Campos et al., 2004*; *Katayama et al., 2011*; *Rasin et al., 2007*), without disrupting the junction integrity of the VZ as revealed by actin filaments (F-actin) (*Figure 1—figure supplement 2*).

Observation in the SVZ and intermediate zone (IZ) of E16.5 mice (electroporated at E13.5) showed that delaminated TBC1D3-expressing cells often exhibited clustered distribution by forming vertical column-like structures (*Figure 1E*), reminiscent of ontogenic radial units (*Rakic, 1988*), expansion of which has been proposed to underlie cortical folding (*Borrell and Gotz, 2014*; *Florio and Huttner, 2014*; *Lui et al., 2011*; *Rakic, 1988*). This column-like aggregation of basal cells induced by TBC1D3 expression may result from increased number of proliferating cells originated

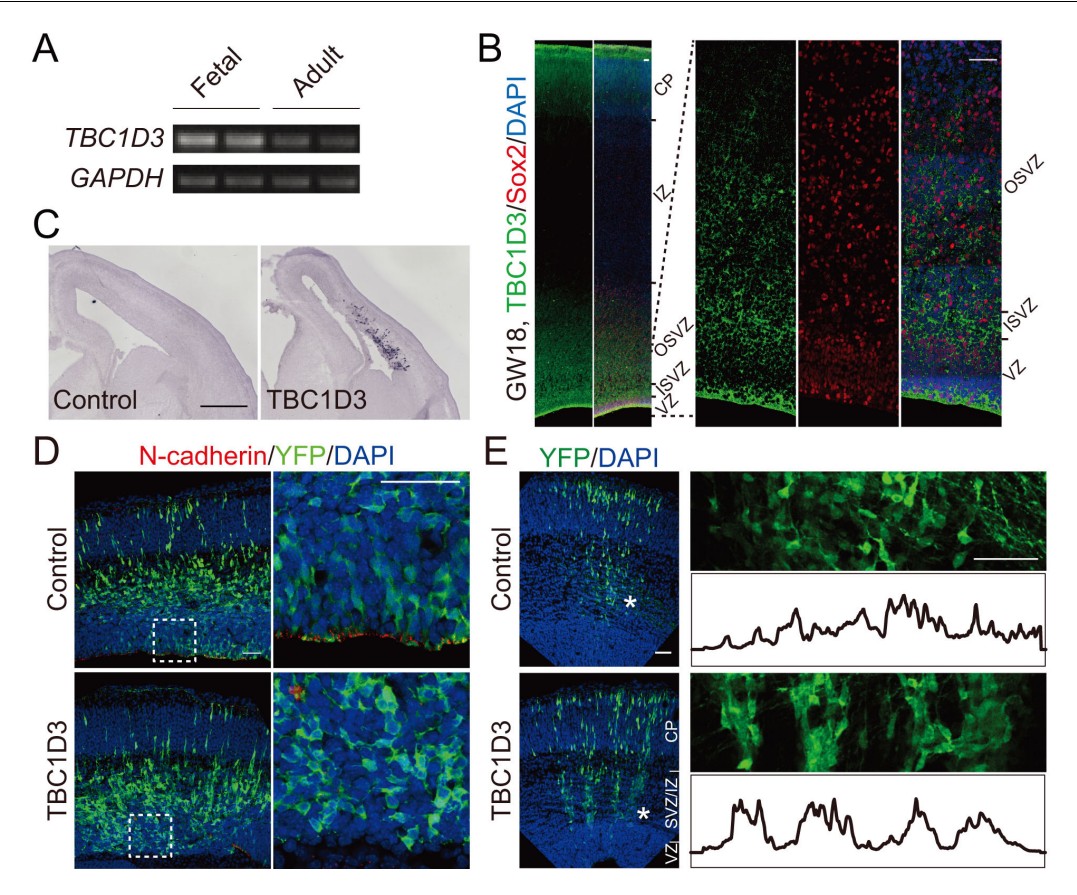

**Figure 1.** TBC1D3 expression in developing mice cortex disrupts adherens junctions and induces formation of column-like structures. (**A**) RT-PCR analysis for the expression of *TBC1D3* mRNA in developing (GW26 - 40) and adult (21–29 years) human whole brain with *GAPDH* as control. (**B**) The expression pattern of TBC1D3 in fetal human cortex at GW18. VZ, ventricular zone; ISVZ, inner subventricular zone; OSVZ, outer subventricular zone; IZ, intermediate zone; CP, cortical plate. Scale bars, 50 µm. (**C–E**) In utero electroporation (IUE) of the mouse cerebral cortex was performed at E13.5 and analyzed at E15.5 (**C** and **D**) or E16.5 (**E**). TBC1D3 or control plasmid was mixed with YFP. (**C**) In situ hybridization analysis for the expression of *TBC1D3* in mouse brain sections. Scale bar, 500 µm. (**D**) Adherens junctions in the ventricular surface were marked by N-cadherin staining. Scale bars, 50 µm. (**E**) Distribution of GFP⁺ cells in E16.5 mice cortex. The right panels show magnified regions indicated by asterisks in left panels, with histograms outlining relative fluorescence intensity of tangentially distributed GFP⁺ cells. Scale bars, 50 µm.

The following figure supplements are available for figure 1:

**Figure supplement 1.** The mRNA level of *N-cadherin* gene is reduced by TBC1D3 expression in mouse cortex.

**Figure supplement 2.** Dislocalization of adherens junctions proteins upon TBC1D3 expression in the VZ.

from VZ (see below), and leads to lateral heterogeneity in the rate of neuronal production that may contribute to cortical folding.

## TBC1D3 expression increases expansion of basal progenitors

Previous studies have shown that disruption of AJs due to the loss of RhoA in neural progenitors is accompanied by an elevated proliferation of neuroprogenitors (*Katayama et al., 2011*). We thus examine the effect of TBC1D3 expression via *in utero* electroporation on neuroprogenitor proliferation by calculating the proportion of cells in S phase, as assayed by the incorporation of pyrimidine analog bromodeoxyuridine (BrdU) for 2 hr before sacrificing the electroporated mice (*Figure 2A,B*). We found that TBC1D3 expression caused an increase in BrdU-positive (BrdU⁺) proliferating cells, as compared to that of control mice electroporated with vehicle construct (*Figure 2C,E*). In an

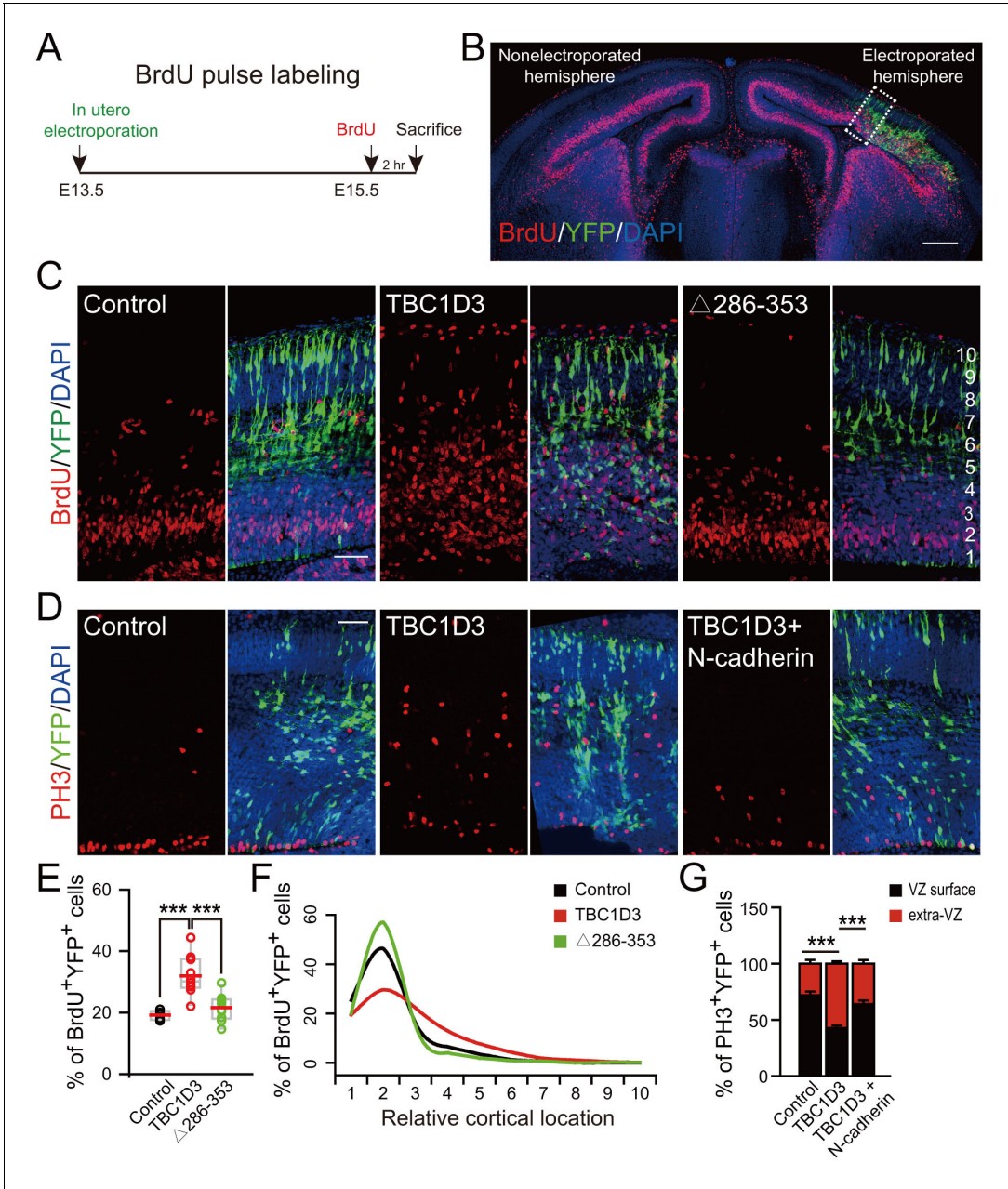

**Figure 2.** Expression of TBC1D3 induces an increase in cell proliferation in basal regions of developing mouse cortex. (**A**) Schematic of BrdU pulse labeling. (**B**) E15.5 whole brain section incorporated with BrdU after IUE at E13.5. Dashed rectangle indicates cortical area for quantification. Scale bar, 200 µm. (**C**) BrdU incorporation in mice subjected to IUE with indicated plasmids. Scale bar, 50 µm. (**D**) Staining for phosphorylated Histone H3 (PH3) in E15.5 mice subjected to IUE at E13.5. Scale bar, 50 µm. (**E**) Quantification for the percentage of BrdU+ cells among electroporated YFP-labeled cells (control: n = 5 mice, mean = 19.18, SEM = 0.69; TBC1D3: n = 10 mice, mean = 31.91, SEM = 2.02; Δ286–353: n = 9 mice, mean = 21.62, SEM = 1.36). Respective p-values are 0.0003 (control vs TBC1D3) and 0.0004 (TBC1D3 vs Δ286–353). (**F**) Mean cortical distribution of BrdU+ cells among electroporated YFP-labeled cells (n = 5 mice). (**G**) Quantification for the percentage of PH3+ cells among electroporated YFP-labeled cells attached to ventricular surface (VZ surface) or out of ventricular zone (extra-VZ). Total number of PH3+ cells in VZ surface and extra-VZ were normalized to 100 for each group (n = 8 mice, mean = 28.10, SEM = 3.43 for extra-VZ of control; n = 10 mice, mean = 57.22, SEM = 2.24 for extra-VZ of TBC1D3; n = 8 mice, mean = 35.99, SEM = 3.26 for extra-VZ of TBC1D3 plus N-cadherin). ***p<0.001.

The following figure supplements are available for figure 2:

**Figure supplement 1.** Colocalization between BrdU and Ki67 in TBC1D3-expressing BPs.

**Figure supplement 2.** Cell autonomous and non-cell autonomous effects of TBC1D3 on neural progenitors.

*Figure 2 continued on next page*

*Figure 2 continued*

**Figure supplement 3.** Detachment of vRGs induced by blocking N-cadherin-mediated adhesion does not promote generation of BPs.

**Figure supplement 4.** Effect of dominant-negative form of Ras on proliferation of TBC1D3-induced BPs.

additional control experiment, we found that expressing a mutated form of TBC1D3, with the deletion of amino acids 286 to 353 ($\Delta$286–353) essential for its cytoplasmic retention (*He et al., 2014*), in neural progenitors had no effect on BrdU incorporation (*Figure 2C,E*). These results suggest that cytoplasmic presence of TBC1D3 promotes the proliferation of early neural precursors. Interestingly, the TBC1D3-induced population of BrdU$^+$ cells scattered widely across the cortex, from VZ to CP, with the highest concentration in SVZ (*Figure 2B,C,F*), indicating increased generation of BPs. The identity of these basal localized cells as BPs was further revealed by positive labeling by Ki67, a marker for cells in cell cycle, and negative labeling by NeuN, a marker for post-mitotic neurons (*Figure 2—figure supplement 1*). This effect of TBC1D3 expression was also confirmed by the increase in the proportion of cells stained with the mitotic marker phosphorylated Histone (PH3$^+$) in basal regions and a decrease in cells attached to the ventricular surface (*Figure 2D,G*). The seemingly non-cell autonomous effect of TBC1D3 expression on proliferation of BPs was most likely caused by different dosages of pE/nestin-TBC1D3 and pCAG-YFP (3:1) used in electroporation, because electroporation with pCAGGS-TBC1D3-IRES-EGFP, a vector co-expressing both TBC1D3 and EGFP, caused increase in numbers of basal BrdU$^+$ or PH3$^+$ cells in EGFP$^+$ but not in EGFP$^-$ cells (*Figure 2—figure supplement 2*). The notable slight decrease of EGFP$^-$ apical neural progenitors might be due to the disruption of proliferation niche in VZ regions (*Figure 2—figure supplement 2B,D*, left).

This increased number of proliferating BPs in basal regions could be caused by increased number of delaminated cells as well as by elevated proliferative capacity of BPs upon TBC1D3 expression. We found that the increase in PH3$^+$ BPs was largely abolished when TBC1D3 and N-cadherin were co-expressed in neuroprogenitors via in utero electroporation (*Figure 2D,G*), consistent with the contribution of delamination to the increased number of proliferating BPs. To determine whether delamination itself is sufficient to transform vRG to BPs, we over-expressed in vRG the extracellular domain (EC1) of N-cadherin, which has been shown to be capable of disrupting the homophilic intercellular N-cadherin interaction (*Tan et al., 2010*), and found that EC1 expression caused detachment of vRG cells (*Figure 2—figure supplement 3A,B*) but had no effect on the proliferation of either apical or basal cells (*Figure 2—figure supplement 3C–F*). Thus, detachment of vRG cells is a necessary but not a sufficient step for the generation of proliferative BPs.

How does TBC1D3 expression maintain the high proliferative capacity of detached RG cells? TBC1D3 has been shown to promote cell proliferation (*Pei et al., 2002*), activate Ras and enhance EGF/EGFR and insulin signaling in non-neuronal cells (*Wainszelbaum et al., 2008*, *2012*). In line with this notion, we found that the effect of TBC1D3 on the proliferation of BPs was markedly abrogated by co-expression with Ras$^{S17N}$ (Ras-DN), a dominant-negative form of Ras (*Figure 2—figure supplement 4*). This result suggests the involvement of Ras signaling in the proliferation of TBC1D3-induced BPs, indicating multiple actions of TBC1D3 in causing increased expansion of BPs.

## Expansion of BPs is caused by elevated oRG proliferation

In mice, intermediate progenitors (IPs) that express the transcription factor Tbr2 in the SVZ are the major type of BPs, which undergo terminal division (*Lui et al., 2011*; *Noctor et al., 2004*). In primates, however, a recently identified subtype of BPs, namely outer radial glia (oRG) or basal radial glia (bRG), represents the predominant BPs in the expanded OSVZ (*Smart et al., 2002*). These oRG cells express RG marker Pax6 and/or Sox2 and can divide multiple times to generate many daughter oRG or IPs (*Fietz et al., 2010*; *Hansen et al., 2010*). By contrast, the lissencephalic mouse embryonic cortex contains very few oRG cells, which divide only once to generate two neurons (*Wang et al., 2011*). Highly proliferative oRG cells are thought to be critical for the expansion of primate brain cerebral cortex (*Hansen et al., 2010*; *Sun and Hevner, 2014*).

Remarkably, we found that TBC1D3 expression caused an increase in Pax6$^+$ cells in the mouse cortex, especially in the basal regions (*Figure 3A–C*). The RG identity of these cells was further corroborated by the oRG-like morphology and the mode of division. In TBC1D3-expressing mouse cortical slices, we observed that many Sox2$^+$ or Pax6$^+$ BPs exhibited a single process pointing to the pial surface with intense signals of mitosis-specific phospho-Vimentin (p-Vim) (*Figure 3—figure supplement 1A–D*), similar to unipolar oRGs identified in the human fetal brain (*Fietz et al., 2010*; *Hansen et al., 2010*). However, in control mice, the percentage of Sox2$^+$/p-Vim$^+$ or Pax6$^+$/p-Vim$^+$ cells with a basal process was much lower (*Figure 3—figure supplement 1A–D*). This pial surface contact by oRG process was further examined by application of fluorescent membrane probe DiI to the isolated brain prior to sectioning. In TBC1D3-expressing cortices, we observed many oRG-like cells with a basal process attaching the pial surface and without the apical process attached to the VZ surface (*Figure 3D,F*). In the control cortices, however, most RGs had both apical and basal processes attaching the pial and VZ surfaces respectively (*Figure 3D,F*). Thus, TBC1D3 expression promoted production of oRG cells, which are normally rather rare in mice. In addition, we observed an increase in Tbr2$^+$ BPs at 96 hr after electroporation with TBC1D3 construct at E13.5 (*Figure 3—figure supplement 1E,F*), in accordance with the linage relationship between oRG cells and IPs (*Hansen et al., 2010*).

To further identify oRG cells among proliferating BPs, we performed time-lapse imaging of 'mitotic somal translocation' (MST) before cytokinesis, a typical oRG behavior during cell division (*Hansen et al., 2010*). We labeled individual progenitors with H2B-GFP to trace nuclear motion and tdTomato to reveal cell morphology without or with co-electroporation with TBC1D3 expression vector pCS2-Myc-TBC1D3, and brains were sliced 24 hr later for time-lapse imaging. In agreement with the above finding of increased oRG population, the number of cells with MST was markedly higher in TBC1D3-expressing slices, as compared to controls (*Figure 3E,G*). To visualize and identify the types of oRGs and their daughter cells generated by TBC1D3 expression, we electroporated E13.5 mice with pCAGGS-TBC1D3-IRES-EGFP plasmids (or pCAGGS-IRES-EGFP as the control), followed with time-lapse imaging (*Videos 1*, *2*) and immunostaining (*Figure 3H–L*) of E14.5 brain slices. More than half (74/120) of proliferating cells labeled by pCAGGS-TBC1D3-IRES-EGFP were typical oRG with a single basal process and an upward MST (*Figure 3H,I*). In agreement with that observed in the primate OSVZ (*Betizeau et al., 2013*), we also observed other types of oRGs, including bipolar cells with both apical and basal processes (42/120) and a few cells with a single apical process and a downward MST (4/120) (*Figure 3H,I*). Post-imaging immunostaining of cell markers showed that, in TBC1D3-electroporated samples, a large fraction of either apical or basal daughter cells derived from oRG cells exhibiting MST continued to express Sox2 but not Tbr2 (*Figure 3J–L*), consistent with oRG cells with the capacity for multiple rounds of cell division. In line with this notion, birth dating analysis using sequential labeling with BrdU and EdU showed that the fraction of BrdU$^+$EdU$^+$ cells was higher in basal regions of TBC1D3-expressing cortices, as compared to that of controls (*Figure 3—figure supplement 2*). Thus, TBC1D3 expression in mice had endowed the primate-like proliferative potency of oRG cells. Taken together, these results show that the primary action of TBC1D3 expression is to promote the generation of oRG cells with high proliferative capacity, leading to expansion of BPs.

## TBC1D3 is essential for the generation of oRGs in cultured human brain slices

To further determine whether TBC1D3 is really essential for the generation of oRG cells in the developing human brain, we took advantage of small interference RNA (siRNA) and investigated the effect of TBC1D3 down-regulation on basal cortical neural progenitors of human brain slices (*Figure 4A–F*). As shown in *Figure 4A*, the constructs encoding TBC1D3 siRNAs (1033, 3B, or 440) down-regulated the expression of TBC1D3 in cultured Hela cells in various degrees with siTBC1D3-440 exhibiting most effective effect. Next, vRG cells in cultured fetal human brain slices were transfected with TBC1D3 or control siRNA using electroporation (*Figure 4B*). After culture for 72 hr, we found that electroporated cells in control samples exhibited normal delamination and a large fraction of them translocated to SVZ regions with typical morphology of oRG cells, whereas majority of siTBC1D3-electroporated cells remained in VZ regions (*Figure 4C,D*). These results suggest that TBC1D3 is critical for the generation of oRG cells. In line with this notion, basal cells positive for Sox2, which largely represents oRG cells (*Pollen et al., 2015*; *Thomsen et al., 2016*), were also

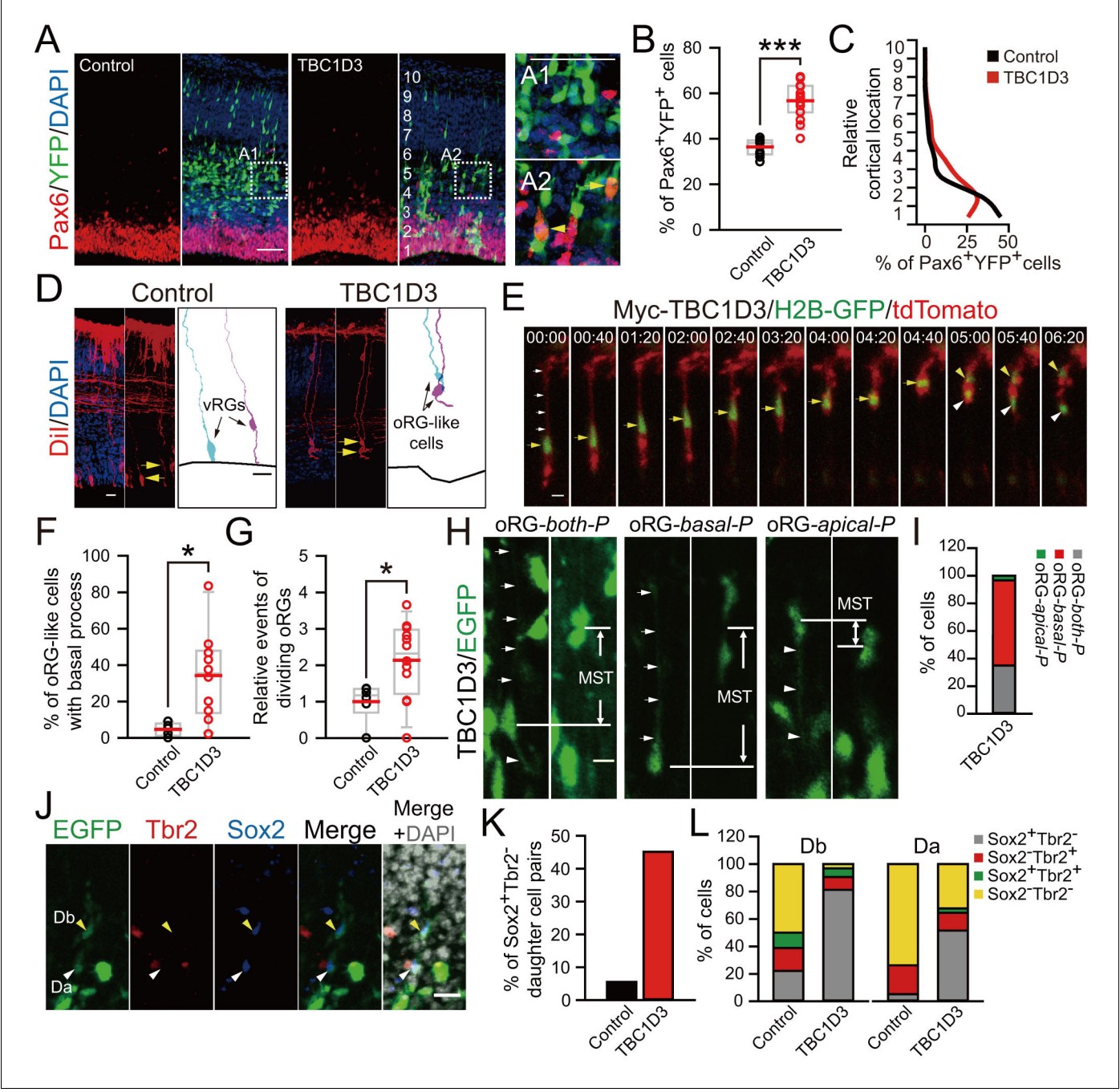

**Figure 3.** TBC1D3 expression in the mouse cortex elevates proliferative oRG cells. (**A**) Staining for Pax6 in E15.5 mice after IUE at E13.5 with pE/nestin-TBC1D3 or pE/nestin (control), together with YFP to mark cell morphology. Scale bars, 50 μm. (**B**) Quantification for the percentage of Pax6+YFP+ cells (control: n = 10 mice, mean = 36.40, SEM = 1.09; TBC1D3: n = 16 mice, mean = 56.68, SEM = 1.97). p<0.0001. (**C**) Mean distribution of Pax6+YFP+ cells. (**D**) DiI-labeling of bipolar ventricular RG cells (vRGs) and oRG-like cells with a basal process attaching the pial surface in E15.5 mouse cortex after IUE at E13.5. Scale bars, 20 μm. (**E**) Time-lapse imaging of TBC1D3-expressing oRG cells (yellow arrows) undergoing division in cultured E14.5 mouse slices after IUE with pCS2-Myc-TBC1D3 and pCAG-H2BGFP-2A-tdTomato at E13.5. H2B-GFP represents cell nucleus. Scale bar, 10 μm. (**F**) Quantification for the percentage of oRG-like cells with a basal process attached to the pial surface and soma located in SVZ or IZ regions, among total RGs including vRGs with soma located in the VZ (control: n = 5 mice, mean = 4.67, SEM = 1.60; TBC1D3: n = 10 mice, mean = 34.21, SEM = 7.56). p = 0.018. (**G**) Quantification for the relative number of basal progenitors with oRG-like divisions, identified by time-lapse imaging in (**E**), per unit of tangential length of the VZ electroporated (control: n = 6 slices, mean = 1.00, SEM = 0.21; TBC1D3: n = 12 slices, mean = 2.14, SEM = 0.30). p = 0.025. (**H** and **I**) Types of TBC1D3-expressing oRG-like cells before division [n = 120 cells in (I)]. Scale bar, 10 μm. (**J**) Immunostaining for Tbr2 and Sox2 in the brain slices after

*Figure 3 continued on next page*

*Figure 3 continued*
time-lapse imaging. Note daughter cells toward basal (Db) or apical (Da) direction upon division. Scale bar, 20 μm. (**K** and **L**) Quantification for
Sox2$^+$Tbr2$^-$ daughter cell pairs (**K**) and daughter cells with indicated marker combinations (**L**) (n = 18 cells for control, n = 31 cells for TBC1D3).

The following figure supplements are available for figure 3:

**Figure supplement 1.** oRG-like cells and IPs increase in the basal region of TBC1D3-expressing mouse cortex.

**Figure supplement 2.** Birth dating analysis for division patterns of neuroprogenitors after IUE.

markedly decreased in siTBC1D3-expressing samples (*Figure 4E,F*). This effect was unlikely caused by off-target effect of siTBC1D3, because co-expression with TBC1D3 largely rescued the defect caused by TBC1D3 down-regulation (*Figure 4E,F*). In agreement with the result that TBC1D3 destabilized *Cdh2* transcript, we found that transfection with TBC1D3 siRNA markedly increased stability of *Cdh2* mRNA in human neural progenitor ReNeuron cells (*Figure 4G*). Taken together, TBC1D3 plays a critical role in the generation of BPs during human brain development.

## TBC1D3 expression induces gyrus-like cortical folding in mice

Although the cortex of small rodents is lissencephalic, it has the potential of forming fissures and folds, as shown in recent studies using regional down-regulation of a DNA-associated protein Trnp1 (*Stahl et al., 2013*) or exogenous application of FGF2 protein in early mouse embryos (*Rash et al., 2013*). We found that about one third of mice electroporated with the TBC1D3 construct in neuroprogenitors at E13.5 showed apparent folding of the cortex at regions containing electroporated cells when observed three days postnatal (P3) (*Figure 5A*). At P7, the cortical folding was more pronounced (*Figure 5A*), suggesting a cumulative effect of TBC1D3 on the formation of gyrus-like cortical folding in mice. The cortical surface 48 hr after electroporation showed continuous signals of laminin in both control and TBC1D3-electroporated samples (*Figure 5—figure supplement 1*), largely excluding the possibility that the later observed cortical folding might be attributed to the disruption of pial basement membrane.

To further determine the effect of TBC1D3 expression on cortical cytoarchitecture, we generated transgenic (TG) mice that expressed TBC1D3 under the control of the *nestin* promoter (*Figure 5—figure supplement 2A*). Three independent founder lines were generated with line 10 exhibiting highest copy number and expression level (*Figure 5—figure supplement 2B,C*), and thus line 10 was used for further breading and analyses. Immunostaining indicated that TBC1D3 is mainly distributed in the cytoplasm of Pax6$^+$ RG cells (*Figure 5—figure supplement 2D*). Notably, the expression level of TBC1D3 was progressively increased from early to later embryonic stages (*Figure 5—figure supplement 2E*). Beginning from E14.5, we observed apparent protrusion of layer 2/3 cells towards the pial surface in the cortex of all TG mice examined (*Figure 5B,C*; arrows), clear folding of cortical surface could be discerned on cortical surfaces or slices with Nissl staining in P3.5 and in adult mice (*Figure 5D–F*; asterisks and arrows). Immunostaining of Cux1 and Ctip2, markers for layer 2/3 and layer 5 cortical neurons, respectively, showed clear lamination in the cortex of both control and TG mice, but the TG cortex showed gyrus-like folding (*Figure 5C,G*). Staining of pan-neuronal marker

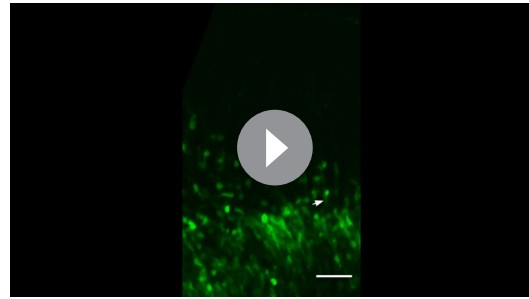

**Video 1.** oRGs in cultured organotypic slices from control mice. This movie illustrates the behavior of the sparse oRGs in organotypic slice cultures from control mice electroporated with vehicle plasmid. IUE was done at E13.5 and organotypic slice culture was prepared at E14.5 and observed 2–3 hr after culture preparation. White arrows indicate oRGs before division, and cyan and red arrowheads indicate daughter cells dividing towards the pial and ventricular surface, respectively. Scale bar, 50 μm.

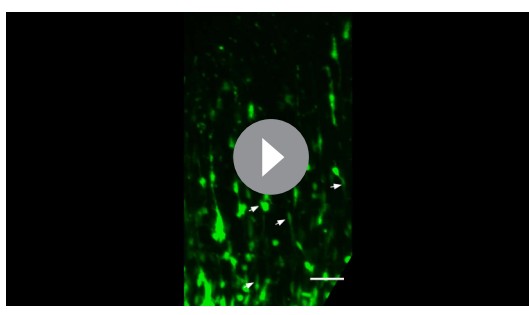

**Video 2.** oRGs in cultured organotypic slices from TBC1D3-electroporated mice. This movie illustrates the behavior of TBC1D3-expressing oRGs. IUE was done at E13.5 and organotypic slice culture prepared at E14.5 and observed 2–3 hr after culture preparation. White arrows indicate oRGs before division, and cyan and red arrowheads indicate daughter cells dividing towards the pial and ventricular surface, respectively. Scale bar, 50 μm.

NeuN and astrocyte marker GFAP also showed folding of the cortical surface in adult TG mice (*Figure 5G,H*). We did not observe apparent difference in vertical distribution of new born cortical neurons in TG mice (*Figure 5—figure supplement 3*), suggesting that cortical folding was unlikely due to neuronal migration defects.

Gross observation showed that all P3.5 TG mice examined exhibited gyrus-like phenotypes in various degrees (*Figure 5—figure supplement 4A*). Analysis for whole-brain serial sections showed that these gyrus-like structures were mainly distributed in the primary (M1) and secondary (M2) motor cortex (*Figure 5—figure supplement 4B–E*). This localized effect may be attributed to temporal restriction of TBC1D3 expression in RG cells of TG mice and the rostral-to-caudal temporal progression of neurogenesis observed in several mammalian species (*Caviness et al., 1995*; *McSherry and Smart, 1986*; *Rakic, 1974*).

Furthermore, we have determined the pattern of neurogenesis in TG mice. At E12.5, the TG mice exhibited an increase in proliferating PH3$^+$ BPs, with a wide distribution across basal regions of the cortex, whereas only a single distinct layer contained PH3$^+$ cells in control mice (*Figure 6—figure supplement 1A–D*). More strikingly, we observed a marked increase in the population of Sox2$^+$Pax6$^+$Tbr2$^-$ cells in basal regions (*Figure 6A, B*) and more than half of them exhibited typical morphology of oRG cells (*Figure 6C*). The presence of elevated oRG cells in the basal regions was also determined by DiI back-labeling on embryos of TG mice (*Figure 6D,E*). Similar to that observed in electroporation experiment (see *Figure 1E*), some of the basal Sox2$^+$Tbr2$^-$ cells exhibited clustered distribution (*Figure 6F,G*). Notably, clustered columns of Sox2$^+$Pax6$^+$Tbr2$^-$ cells were observed beneath regions of protrusion (as defined by DAPI staining of cells) at E14.5 (*Figure 6H*), supporting the ontogeny unit hypothesis for cortex expansion (*Rakic, 1988*). Unlike basal Pax6$^+$ cells, relative density of total Pax6$^+$ only slightly increased (*Figure 6—figure supplement 1E,F*). Sequential labeling with BrdU and EdU for mice at E13.5 and E16.5 also showed an increase in the number of EdU$^+$BrdU$^+$ basal progenitors in TG mice (*Figure 6—figure supplement 2*). These results further support the conclusion based on *in utero* electroporation studies that TBC1D3 expression promotes the generation of highly proliferative BPs in mice. Besides morphological features and mode of division, several molecular markers have been identified for oRG cells, including HOPX (*Pollen et al., 2015*; *Thomsen et al., 2016*). Interestingly, many HOPX$^+$ cells were observed in basal extra-VZ regions of TG mice, whereas these cells were barely detectable in WT control mice (*Figure 6I,J*). Thus, we conclude that TBC1D3 expression promotes generation of oRG cells in mice.

We next examined whether neuronal density in the cortex is affected in TG mice expressing TBC1D3. Brain sections were immunostained with Cux1/Ctip2 or NeuN/GFAP. At both P3.5 and P28, TG mice exhibited increased neuronal density in the motor cortex showing surface folding, particularly in the layer 2/3 (Cux1$^+$), as compared to corresponding regions in WT littermates (*Figure 6—figure supplement 3A–E*). In adult mice (P28), we did not observe apparent changes in the density of GFAP$^+$ astrocytes (*Figure 6—figure supplement 3C,F*). These results suggest that TBC1D3-induced cortical expansion is mainly due to increased number of neurons rather than astrocytes. Although cortical folding was enriched in motor cortex, we also observed a mild increase in the density of Cux1$^+$ in sensory cortex, which formed much less folds (*Figure 5—figure supplement 4* and *Figure 6—figure supplement 3G,H*).

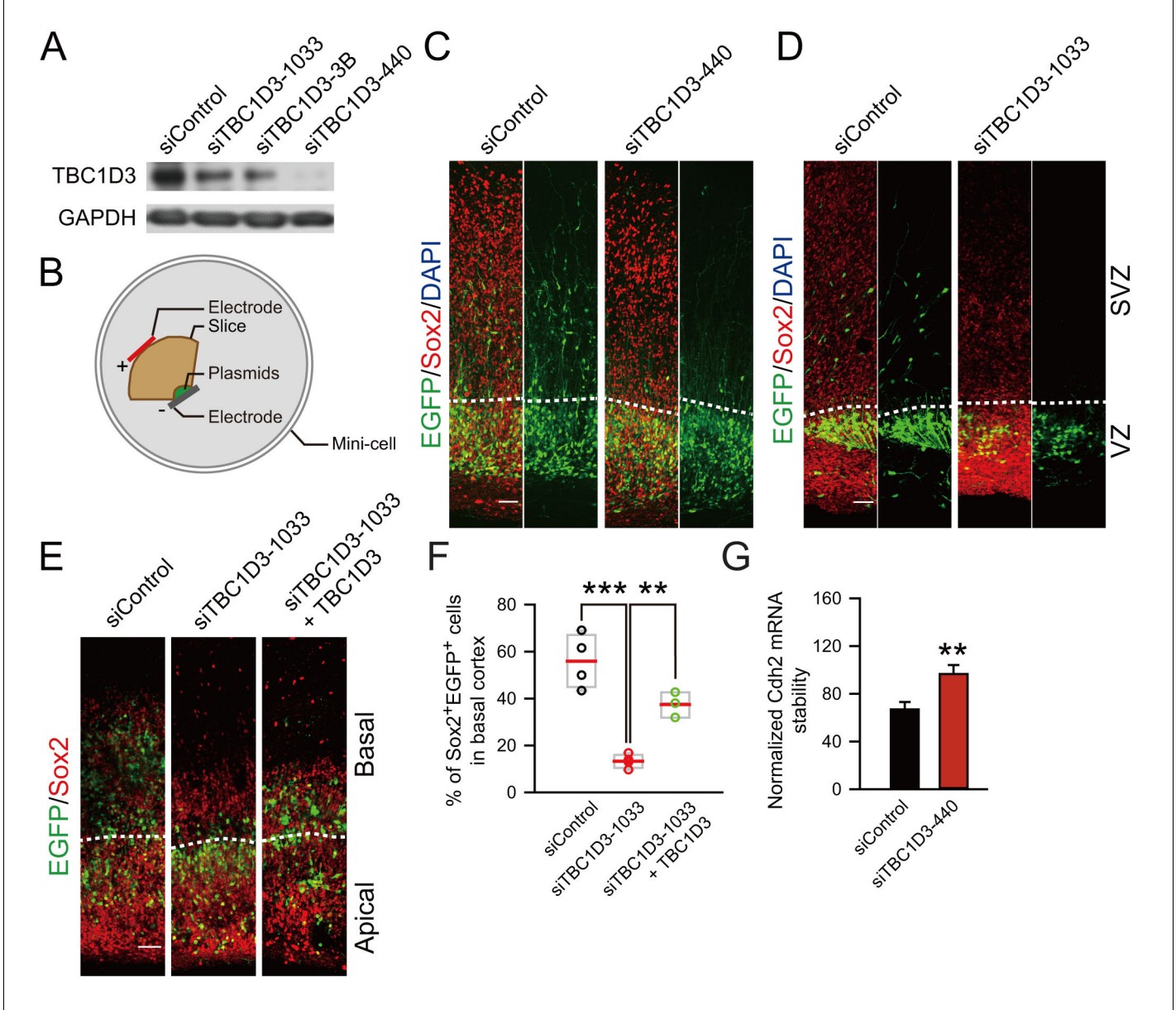

**Figure 4.** Knockdown of TBC1D3 in human vRGs inhibits the generation of oRGs. (**A**) Detection of TBC1D3 protein levels in Hela cells transfected with pSuper-siTBC1D3 plasmids, with a scramble sequence as the control. (**B**) Paradigm of culture and electroporation of human brain slice. (**C–E**) The VZ of human brain slices at GW14.5 (**C**), GW17.1 (**D**), GW13.5 (**E**) were transfected with pSuper-siTBC1D3 plasmids or a plasmid encoding scrambled sequence as the control, without or with co-transfection with TBC1D3 expression plasmid (pCS2-Myc-TBC1D3) by electroporation method as described in (**B**), followed by staining with Sox2 antibody at 72 hr post electroporation. Scale bars, 50 µm. (**F**) Quantification for the percentage of Sox2$^+$ cells among total EGFP$^+$ cells in basal regions (control: n = 4 slices, mean = 55.91, SEM = 5.76; siTBC1D3: n = 4 slices, mean = 13.32, SEM = 1.46; siTBC1D3 plus TBC1D3: n = 3 slices, mean = 37.47, SEM = 3.12). p = 0.0002, control vs siTBC1D3; p = 0.004, siTBC1D3 vs siTBC1D3 plus TBC1D3. (**G**) Human ReNeuron cells were transfected with siTBC1D3 or control plasmid for 3 days followed by treatment with actinomycin D for 4 hr. The mRNA levels of *Cdh2* in ReNeuron cells after actinomycin D treatment were quantified (control: n = 6 experiments; mean = 66.99, SEM = 6.21; siTBC1D3: mean = 96.62, SEM = 7.62; p = 0.003), normalized to that in cells with 0 hr of actinomycin D treatment.

## TBC1D3 expression down-regulates Trnp1 transcription and up-regulates ERK signaling

Because delamination and elevation of the proliferative capacity of BPs are both required to account for the observed TBC1D3-induced oRG generation and cortical folding, we further examined the

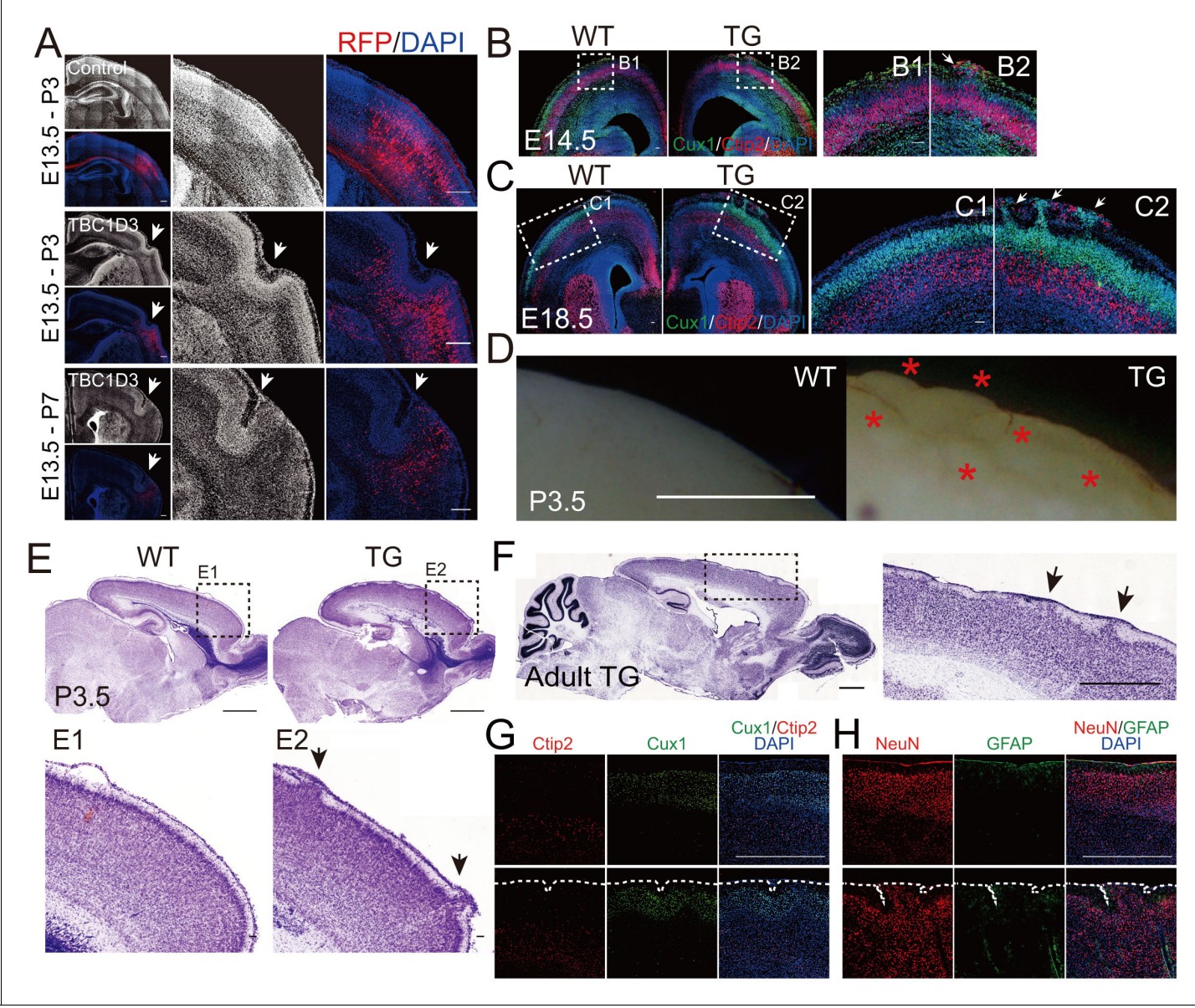

**Figure 5.** Cortical folding and gyrification induced by TBC1D3 expression in mice. (**A**) E13.5 mice were subjected to IUE with pE/nestin-TBC1D3 or vehicle control plasmids, together with RFP to mark electroporated cells, and analyzed at the indicated time. White arrows indicate cortical folds. Scale bars, 200 μm. (**B** and **C**) Slices from WT or TG mice at E14.5 (**B**) or E18.5 (**C**) were stained for Cux1 and Ctip2. Note the protrusions (white arrows) in TG mice (B2 and C2) compared to smooth surfaces in WT mice (B1 and C1). Scale bars, 50 μm. (**D**) Images from whole mount P3.5 WT and TG brains. Note the convoluted cortical surfaces indicated by red asterisks in TG mice. Scale bar, 1 mm. (**E** and **F**) Nissl staining of sagittal sections of TG or WT mice at P3.5 (**E**) or adult stage (F, 3 months). Note the apparent gyrus-like structures (black arrows) in boxed areas. Scale bars, 1 mm (**E** and **F**) or 50 μm (E1 and E2). (**G** and **H**) Immunostaining for Cux1 and Ctip2 (**G**) or NeuN and GFAP (**H**) in the adult WT and TG mouse brain sections. Note the folded cortical surfaces outlined by dash lines; Scale bars, 1 mm.

The following figure supplements are available for figure 5:

**Figure supplement 1.** TBC1D3 electroporation has no effect on the pial basement membrane integrity.

**Figure supplement 2.** Generation of TBC1D3 transgenic mouse.

**Figure supplement 3.** Normal neuronal migration in the cortical plate of TG mice.

**Figure supplement 4.** Cortical folding mainly occurs in the motor cortex of TBC1D3-transgenic mice.

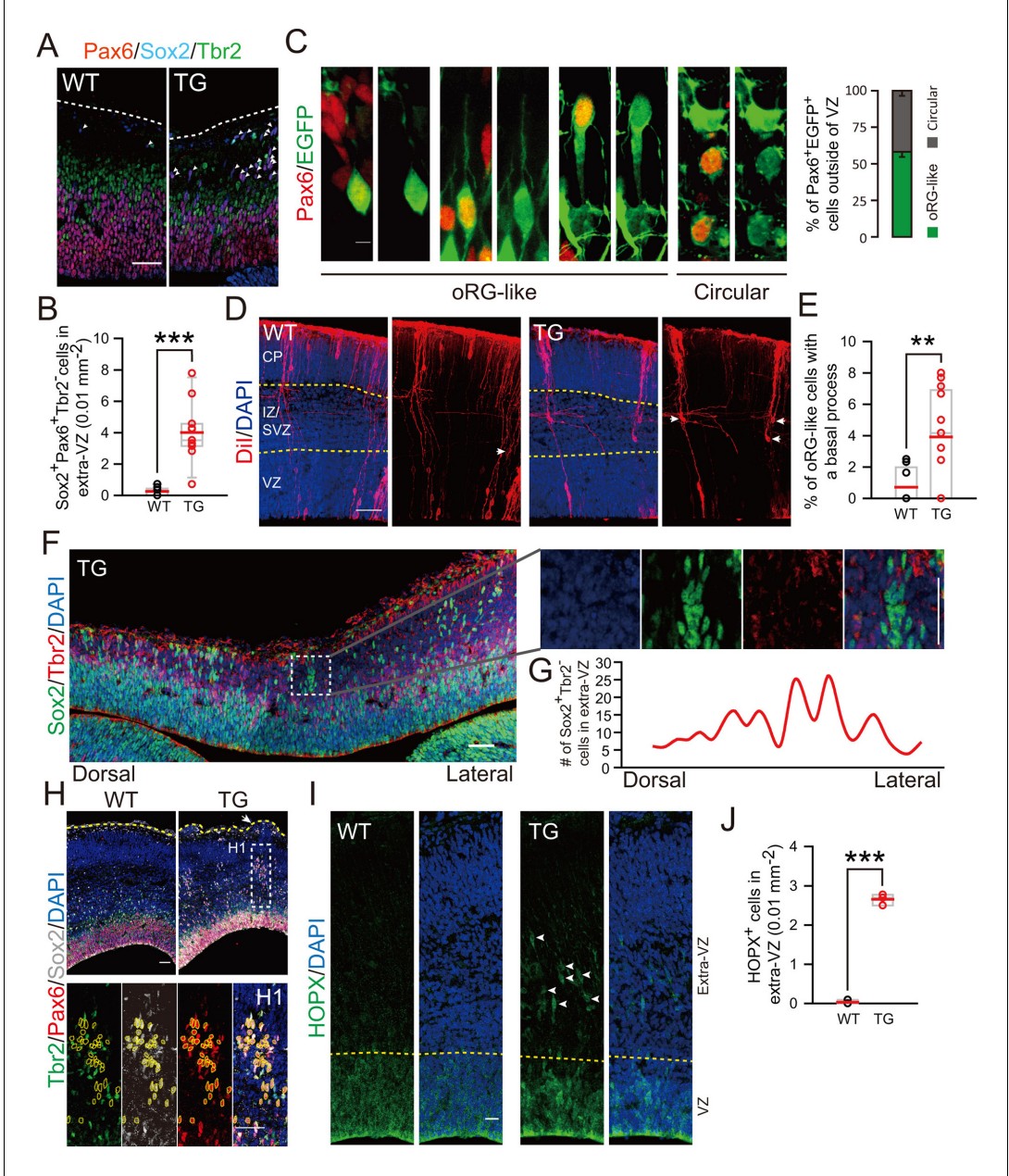

**Figure 6.** Cortical basal progenitors are increased and display columnar distribution in the cortex of TBC1D3-transgenic mice. (A) Staining for Pax6, Sox2, and Tbr2 in E12.5 WT and TG mice. Dash lines indicate pial surfaces. Note the increase in Pax6+Sox2+Tbr2- cells (white arrowheads) in the basal region of TG mice. Scale bar, 50 μm. (B) Quantification for the density of Pax6+Sox2+Tbr2- cells in extra-VZ regions (WT: n = 9 slices from 3 mice, mean = 0.26, SEM = 0.09; TG: n = 11 slices from 4 mice, mean = 4.01, SEM = 0.57). p<0.0001. (C) E13.5 TG mice were subjected to IUE with EGFP-expressing plasmids to label cell morphology, and brain sections were stained for Pax6 at E16.5 (left panel). Scale bar, 5 μm. Three types of oRG-like cells constitute more than half of Pax6+ BPs (right panel, 82 Pax6+EGFP+ cells from 5 brains were analyzed). (D and E) DiI labeling (D) and quantification (E) of RG cells in E14.5 WT and TG mice (WT: n = 9 slices, mean = 0.72, SEM = 0.37; TG: n = 13 slices, mean = 3.92, SEM = 0.88). p = 0.009. White arrows indicate typical oRG-like cells with soma located in the SVZ/IZ and a basal process attached to the pial surface. Scale bar, 50 μm. (F) E12.5 TG mice were stained with Sox2, and Tbr2. Note the columnar distribution of Sox2+Tbr2- cells in basal regions, as illustrated for the boxed area. Scale bar, 50 μm. (G) Distribution profile of Sox2+Tbr2- cells in the basal region of TG mice cortex ranging from dorsal to lateral cortical regions. (H) Immunostaining for Sox2, Pax6, Tbr2 in E14.5 WT and TG mice cortex. Yellow dash lines indicate the brain surface. Note the apparent columnar distribution of Sox2+Pax6+Tbr2- cells (yellow dotted circles) in the boxed area below a cortical gyrus-like structure (white arrow). Scale bars, 50 μm. (I and J) Immunostaining (I) and quantification (J) of HOPX cells in the extra-VZ (white arrowheads) of E14.5 WT and TG mice cortices (WT: n = 3 brains, mean = 0.03, SEM = 0.03; TG: n = 3 brains, mean = 2.66, SEM = 0.08). p<0.0001. Scale bar, 20 μm.

*Figure 6 continued on next page*

*Figure 6 continued*

The following figure supplements are available for figure 6:

**Figure supplement 1.** Cortical basal progenitors are increased in the cortex of TBC1D3-transgenic mice.
**Figure supplement 2.** Increased proliferation potency of BPs in TBC1D3-transgenic Mice.
**Figure supplement 3.** Increased neurons in the superficial layer of the cortex of TBC1D3-transgenic mice.

effect of TBC1D3 expression on the intrinsic stemness signaling pathways in BPs. Interestingly, we found that TBC1D3 expression in neuroprogenitors by in utero electroporation at E13.5 caused a reduced expression of *Trnp1* in flow cytometry-sorted cells at E15.5, as compared to that found in control cells sorted from vehicle-electroporated mice (*Figure 7A*), consistent with the finding that regional down-regulation of *Trnp1* expression causes cortical folding in mice (*Stahl et al., 2013*). Unlike the role in destabilizing *Cdh2* transcript, TBC1D3 expression had no effect on the stability of *Trnp1* mRNA (*Figure 7E*). This result indicates a linkage between TBC1D3 and known factors that modulate fate transition from RGs to BPs. In addition to *Trnp1*, Notch signaling has been proposed to play a role in maintaining the progenitor status of human oRG cells (*Hansen et al., 2010*). However, we found that TBC1D3 expression had no effect on the transcriptional level of *Hes1* and *Hes5*, effecters of Notch signaling, as well as *Numb*, which encodes an endocytic adaptor protein that acts as a Notch pathway inhibitor localized to the apical membrane (*Figure 7B–D*). The Ras-Raf-ERK signaling cascade mediates the mitotic role of EGF/EGFR signaling pathway in promoting cell proliferation (*Citri and Yarden, 2006*). We found that the signals for activated phospho-ERK1/2 (pERK1/2) were elevated in TBC1D3 TG mice, mostly in VZ/SVZ regions (*Figure 7F,G*), in line with the previous observation that TBC1D3 enhanced the ERK signaling (*Wainszelbaum et al., 2008*, *2012*). Notably, almost all (99.1%) of $Pax6^+Tbr2^-$ oRG cells in developing human cortex were positively stained by pERK1/2 (*Figure 7H,I*), further supporting the role of ERK signaling in oRG proliferation. The combined actions of TBC1D3 on parallel or separate cell proliferation pathways may account for the observed role of TBC1D3 in maintaining cell stemness potency.

## Discussion

In this study, we found a remarkable effect of the hominoid-specific gene TBC1D3 in promoting the generation of oRG cells in mice with a high proliferative ability previously observed in primates. Importantly, this oRG generation is accompanied by the appearance of cortical folding. Furthermore, we elucidated the cellular mechanisms underlying the generation of proliferative oRG cells by showing that TBC1D3 expresssion caused increased delamination of VZ neuroprogenitors via down-regulation of N-cadherin, and elevated proliferative ability of BPs is accompanied by a reduced expression of *Trnp1* and likely involves Ras-ERK signaling cascade. These effects together resulted in the expansion and dispersion of BPs, giving rise to increased number of newborn neurons and leading to cortical folding (for model see *Figure 7J*).

In small rodents, RGs in the VZ and IPs in the SVZ are two major cell types responsible for cortex development. However, amplification of VZ progenitors via overexpression of β-catenin in the mouse brain led to the folding of the ventricular rather than cortical surface (*Chenn and Walsh, 2002*), and amplification of IPs only increased the brain size without inducing cortical folding in mice (*Nonaka-Kinoshita et al., 2013*). Recent studies have established the association between the relative abundance of oRG and the degree of cortical folding in various species (*Betizeau et al., 2013*; *Hansen et al., 2010*; *Reillo et al., 2011*; *Shitamukai et al., 2011*; *Wang et al., 2011*). In the present study, we showed that increased population of oRG in mice is accompanied with cortical fold formation, supporting the critical role of highly proliferative oRG in cortical expansion and gyrification in primates.

It is postulated that oRG cells are derived from the ventricular RGs, presumably requiring delamination from apical anchoring (*Borrell and Gotz, 2014*). However, loss of adherens junction proteins or breaking their linkage to the cytoskeletal belts by down-regulating the small GTPase RhoA (*Cappello et al., 2012*; *Lien et al., 2006*) had little effect on oRG generation or cortical folding,

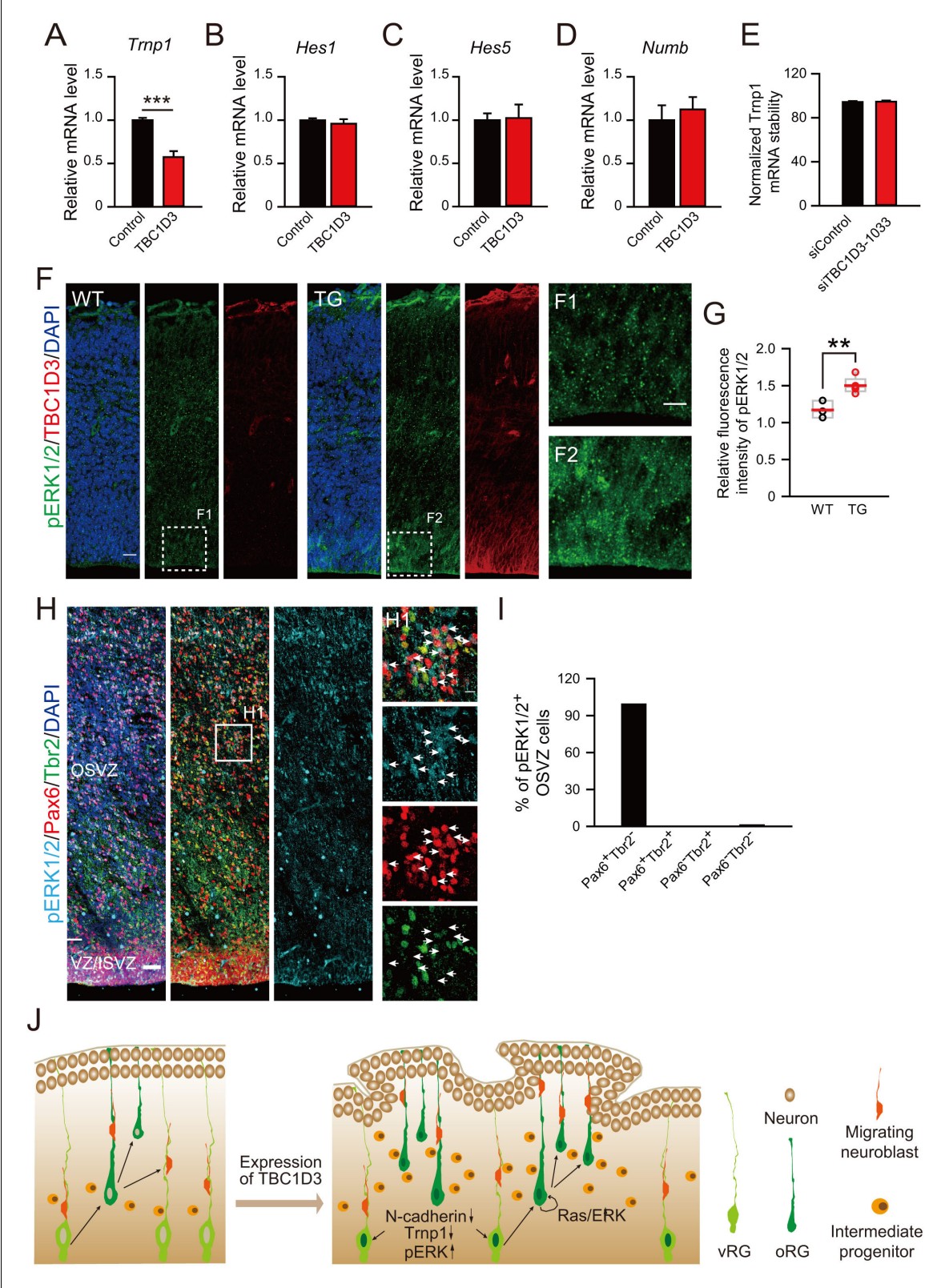

**Figure 7.** Effect of TBC1D3 on intrinsic stemness signaling pathways and model for the TBC1D3 function in cortical folding. (A–D) E13.5 mice were subjected to IUE with TBC1D3 or vehicle control plasmid, together with YFP plasmid, and at E15.5, FACS-sorted transfected cells (see *Figure 1—figure supplement 1B,C*) were analyzed for mRNA levels of indicated genes relative to *GAPDH* with values from control groups normalized as 1.0 (n = 3 experiments for each group). *Trnp1*: mean = 1.00, SEM = 0.03 for control; mean = 0.57, SEM = 0.07 for TBC1D3 (p = 0.0003). *Hes1*: mean = 1.00, SEM = *Figure 7 continued on next page*

*Figure 7 continued*

0.02 for control; mean = 0.96, SEM = 0.05 for TBC1D3 (p = 0.561). *Hes5*: mean = 1.00, SEM = 0.20 for control; mean = 0.85, SEM = 0.34 for TBC1D3 (p = 0.762). *Numb*: mean = 1.00, SEM = 0.17 for control; mean = 1.12, SEM = 0.14 for TBC1D3 (p = 0.612). (E) ReNeuron cells were transfected with constructs encoding siRNA targeting TBC1D3 or scrambled sequence. After 3 days, transfected cells were treated with actinomycin D for 4 hr and the mRNA level of *Trnp1* relative to *Hprt* was quantified (n = 4 experiments; mean = 94.40, SEM =0.89 for control; mean = 94.67, SEM = 1.05 for siTBC1D3; p = 0.851). Data are presented as mean ± SEM of percentage of *Trnp1* mRNA compared to the value prior to actinomycin D treatment. (F) Cortices of E14.5 WT and TG mice were stained for pERK1/2 and TBC1D3. Scale bars, 20 μm (F), 10 μm (F1). (G) Quantification for the ratio of pERK1/2 intensity in VZ/SVZ regions to that in CP (WT: n = 3 mice, mean = 1.17, SEM = 0.07; TG: n = 5 mice, mean = 1.50, SEM = 0.05). p = 0.006. (H) GW15.5 human brain slice was stained with pERK1/2, Pax6, and Tbr2 antibodies. Scale bars, 50 μm (H) and 10 μm (H1). (I) Quantification of pERK1/2 levels in different types of OSVZ progenitors (n = 2 slices). Note that almost all pERK1/2 signals are detected in Pax6$^+$Tbr2$^-$ cells. (J) Proposed model for the role of TBC1D3 in cortical folding. TBC1D3 expression causes delamination of vRG cells, through down-regulating the level of N-cadherin and Trnp1, and promotes proliferation of oRG-like cells by regulating cell stemness pathways, including Ras/ERK signaling. The increased generation of the oRG-like cells, the IP cells, and subsequently regional increase in the density of new born neurons, induces cortical folding in mice.

although apically anchored progenitor cells became delaminated. Thus, delamination of VZ progenitors is not sufficient for generating highly proliferative BPs, and changes in cell cycle regulatory factors are also be required. In the present study, we showed that expression of TBC1D3 led to not only down-regulation of N-cadherin, leading to VZ progenitor delamination, but also reduced Trnp1 expression and enhanced ERK signaling that could be involved in elevated proliferation of delaminated cells. Furthermore, other functions of TBC1D3 may also contribute to the cortical folding phenotype we observed. For examples, cell culture studies of TBC1D3 have suggested its potential role in membrane endocytosis, as well as Ras activation and epidermal growth factor receptor (EGFR) signaling (*Wainszelbaum et al., 2008*), processes that may be involved in various aspects of neural development leading to cortical folding and gyrification. Indeed, we found that Ras activity was involved in TBC1D3-induced proliferation of BPs. Recent single-cell transcriptome analyses of human oRG cells have identified multiple preferentially expressed genes related to extracellular matrix formation, cell migration, and stemness, including the most specific oRG marker HOPX (*Pollen et al., 2015*; *Thomsen et al., 2016*). Strikingly, many HOPX positive cells were observed in TBC1D3 TG mice. The relationship between TBC1D3 expression and these oRG markers remains to be clarified. Simple ectopic localization of RG cells outside of the ventricular zone may not be sufficient to cause cortical expansion or folding as shown in a previous study (*Yoon et al., 2014*). Appearance of oRG-like features, including the contact with pia surface by the basal process, multiple rounds of cell division, expression of specific markers, as well as clustered regional distribution of BPs observed in this study, may play prominent roles in cortical folding.

Cortical expansion is assumed to be associated with the emergence of uniquely cognitive skills in primates (*Luders et al., 2008*; *Zilles et al., 2013*). However, this hypothesis has been mainly based on across species comparative studies. Global or regional-specific changes in gyrification index (GI) have been observed in brains of subjects following various types of extensive training (*Amunts et al., 1997*; *Luders et al., 2012*; *Luders et al., 2008*). For example, in keyboard players, the local GI increases in response to early onset of professional training and duration of practice, and correlates with motor performance (*Amunts et al., 1997*). Conversely, a significant decline in the cortical GI was observed in patients with mental disorders (*Bonnici et al., 2007*; *Wolosin et al., 2009*). Understanding of the precise contribution of cortical expansion and gyrification to cognition functions has been challenging due to the lack of appropriate animal model systems. The deliberate behavior analyses for TG mice generated in this work may provide a link between cortex expansion and higher brain functions.

Recently, a human-specific gene *ARHGAP11B* has been shown to promote production of basal IPs and cause cortical folding in the electroporated mouse brain (*Florio et al., 2015*). We propose that *TBC1D3*, which appeared earlier than *ARHGAP11B* during human evolution (*Florio et al., 2015*; *Hodzic et al., 2006*; *Perry et al., 2008*), acts as a candidate gene that controls many other genes involved in cortex expansion and folding. Multiple genes underlying different or shared cellular processes associated with neurogenesis may all contribute in part to the formation of cortical folding and gyrification found in primates. Thus, it would be of great interest to elucidate gene expression networks related with TBC1D3 expression. Identification of the 'hub' genes of such networks may

further clarify the role of TBC1D3 in the cytoarchitectural development on the cortex. This study provides proof-of-concept that the brain development can be regulated by duplicated genes during hominoid evolution.

## Materials and methods

### Human fetal brain collection
Human fetal brain tissue samples were collected at autopsy within 3 hr of spontaneous abortion with the informed consent of the patients following protocols and institutional ethic guidelines stated in our previous study (*Ma et al., 2013*). Brain tissues were stored in ice-cold Leibowitz-15 medium and transported to the laboratory for further examination and processing (*Lui et al., 2014*).

### Animals
ICR mice were used for all in utero electroporation experiments, and the TBC1D3-transgenic mouse was constructed and kept in the C57BL/6 background. All the mice were housed in the institutional animal care facility with a 12 hr light-dark schedule. The use of all mice in this study was in compliance with the guidelines of the Institutional Animal Care and Use Committee.

### Analysis for TBC1D3 mRNA levels in human samples
Fetal (GW 26 to 40) or adult (21 to 29 years) human brain RNA samples (Clontech, Mountain View, CA) were reverse transcribed into cDNA with QuantScript RT kit using oligo $(dT)_{15}$ primers (Tiangen, China) followed by PCR using TBC1D3 primers: 5'- ATCGAGCGTACAAGGGAATG-3' (forward), 5'–CCGTATCGATCCCTGAAGAA-3' (reverse). GAPDH was used as the control.

### Plasmids and *in u*tero electroporation
Fetal human brain RNA (Clontech) was reverse transcribed with QuantScript RT kit using oligo $(dT)_{15}$ primers (Tiangen). TBC1D3 cDNA was acquired by PCR and cloned into pE/nestin-EGFP vectors (a kind gift from Dr. H. Okano) (*Kawaguchi et al., 2001*), to generate the pE/nestin-TBC1D3 plasmid after replacing the EGFP sequence. To construct the pCAGGS-TBC1D3-IRES-EGFP plasmid, TBC1D3 cDNA was amplified by PCR from pE/nestin-TBC1D3 and subcloned into the AscI-XhoI site of pCAGGS-IRES-EGFP. The pE/nestin-TBC1D3$^{\Delta286-353}$ ($\Delta286-353$) plasmid was produced by site-directed PCR mutagenesis method. The pCS2-Myc-TBC1D3 construct was generated by inserting TBC1D3 cDNA fragment into the EcoRI-XhoI site of pCS2-MT vector. The pKH3-N-cadherin and pCS2-EC1 (*Tan et al., 2010*) plasmids were kind gifts from Dr. X. Yu. Following oligonucleotides were synthesized (Invitrogen, Waltham, MA) for the generation of vectors encoding small interference RNA targeting TBC1D3: siTBC1D3-1033 (forward, 5'- GATCCCCGCCTCTATGAAGAAACTAATTCAAGAGATTAGTTTCTTCATAGAGGCTTTTTGGAAA -3', reverse, 5'- AGCTTTTCCAAAAAGCCTCTATGAAGAAACTAATCTCTTGAATTAGTTTCTTCATAGAGGCGGG -3'), siTBC1D3-440 (forward, 5'- GATCCCCGGGACGTAAGCGGGACATTAATTCAAGAGATTAATGTCCCGCTTACGTCCCTTTTTGGAAA -3', reverse, 5'- AGCTTTTCCAAAAAGGGACGTAAGCGGGACATTAATCTCTTGAATTAATGTCCCGCTTACGTCCCGGG -3'); siTBC1D3-3B forward, 5'- GATCCCCGGATATTGATTGACGGGATTTCAAGAGAATCCCGTCAATCAATATCCTTTTTGGAAA -3', reverse, 5'- AGCTTTTCCAAAAAGGATATTGATTGACGGGATTCTCTTGAAATCCCGTCAATCAATATCCGGG -3') (*Frittoli et al., 2008*). After annealing, the oligos were inserted into BglII/HindIII digested pSuper plasmid. *In utero* electroporation was performed according to the previously reported protocol (*Saito, 2006*). In brief, a timed pregnant mouse at E13.5 was anesthetized with pentobarbital sodium, the uterine horns were exposed, and ~1 µl of plasmids mixed with 0.1 mg/ml fast green (Sigma-Aldrich, St. Louis, MO) were manually microinjected into the lateral ventricle with a beveled sharp glass micropipette (VWR International, Radnor, PA). For electroporation, five 50-ms pulses of 35 mV with a 950-ms interval were applied across the uterus with two 3-mm, in radius, disc electrodes (BEX, Japan, LF650P3) located on either side of the head (BTX, Holliston, MA, ECM830) and the pre-warmed (37°C) 0.9% NaCl was used to keep the uterus wet. After electroporation, the uterus was put back into the abdominal cavity, filled with warmed 0.9% NaCl, and the wound was surgically sutured. The mouse was then placed on a warmed blanket before recovery and resuming normal activity.

## mRNA stability assay

The human neural progenitor cell line ReNeuron was cultured on 0.5% (v/v) of 1 mg/ml laminin-coated dishes in DMEM/F12 full media containing 2% B27 (Gibco, Holliston, MA), 10 units/ml heparin (Sigma), 20 ng/ml EGF (Millipore, Temecula, CA), and 10 ng/ml bFGF (Millipore). For plasmid transfection, the cultured ReNeuron cells were dissociated into single cells with accutase (Sigma) and electroporated in the program X-001 by using the Nucleofector 2b device (Lonza, Switzerland), with plasmids encoding TBC1D3, siRNA against TBC1D3, or corresponding vehicle plasmids. At the second (for TBC1D3 over-expression) or third day (for TBC1D3 knockdown) post transfection, cells were treated with 1 µg/ml actinomycin D (Sigma) for 0, 2 or 4 hr to inhibit gene transcription. Total RNA was extracted with Trizol (Ambion, Holliston, MA) and analyzed by RT-qPCR to detect the respective mRNA levels with following primers: human *Cdh2* (forward, 5′ -ATGAAAGACCCA TCCCACG- 3′, reverse, 5′-TCCTGCTCACCACCACTA-3′); human *Fos* (forward, 5′-TCCGAAGG-GAAAGGAATAA-3′, reverse, 5′-TGAGCTGCCAGGATGAACT-3′); human *Hprt* (forward, 5′-TGACC TGCTGGATTACAT-3′, reverse, 5′-TTGGATTATACTGCCTGA-3′); human *TBC1D3* (forward, 5′-AGG TTCAGCAGAAGCGCCTCA-3′, reverse 5′-GCCTGGATGCCGACGACCCTT-3′); human *Trnp1* (forward, 5′-GGAGGGGACGGCAGAGCAGA-3′, reverse 5′-GGGTCGGGGTAGGAGTCAAGGT-3′). The vitality and transfection efficiency of ReNeuron cells were monitored under fluorescence microscopy during actinomycin D treatment and before RNA extraction.

## Histology, immunohistochemistry and confocal imaging

Postnatal mouse was perfused with phosphate buffered saline (PBS) followed with cold 4% paraformaldehyde (PFA) and prenatal mouse was perfused with cold 4% PFA directly, and then the brain was dissected out and post-fixed into cold 4% PFA in PBS at 4°C overnight. The fixed brain was dehydrated in 20% sucrose in PBS at 4°C and ultimately sectioned into 30 µm cryosections collected on glass slides, or 50 µm cryosections floating in PBS. The fetal human brain tissues were fixed in 4% PFA in PBS at 4°C for 3 days and dehydrated in 30% sucrose in PBS. After embedded and frozen at −80°C in O.C.T. compound (Tissue-Tek, Hatfield, PA), the tissues were sectioned into 40 µm cryosections and stored at –80°C. For histological analysis, frozen sections were stained with 1% cresyl violet (Sigma) for *Nissl* staining. For immunohistochemistry, mouse brain slices were washed in PBS for 3 times and permeated in 0.3% (v/v) Triton X-100 in PBS for 30 min at room temperature (RT). After above treatments, brain sections were incubated directly in a blocking solution (10% (v/v) donkey serum in PBS) for 1 hr, followed by the incubation with the primary antibodies at 4°C overnight. Sections were then washed with PBS for 3 times followed by incubation with the appropriate secondary antibodies for 1–2 hr at room temperature (RT). For the labeling of actin filaments, mouse brain slices were incubated with phalloidin-Alexa 647 (1:40, Invitrogen) in blocking solution with 0.1% Triton X-100 for 1–2 hr at RT. Fetal human brain cyrosections were subjected to heat-induced antigen retrieval in 10 mM sodium citrate (pH = 6.0) for 10 min, then followed by above procedures. All labeled sections were mounted with fluorescent mounting medium (Dako, Carpinteria, CA) and stored at 4°C. The primary antibodies used were: mouse anti-TBC1D3 (Santa Cruz, Dallas, TX, sc-376073, 1:100), mouse anti-NeuN (Millipore MAB377, 1:500), rat anti-Ctip2 (Abcam, Cambridge, MA, ab18465, 1:1,000), rabbit anti-Cux1 (Santa Cruz sc-13024, 1:200), rabbit anti-GFP (Invitrogen A11122, 1:1,000), chicken anti-GFP (Aves Lab, Tigard, OR, GFP-1020, 1:500), goat anti-Sox2 (Santa Cruz sc-17320, 1:200), rabbit anti-Sox2 (Millipore ab5603, 1:500), rabbit anti-Pax6 (Covance, Princeton, NJ, PRB-278P, 1:1,000), rabbit anti-Tbr2 (Abcam ab23345, 1:500), chicken anti-Tbr2 (Millipore AB15894, 1:200), mouse anti-phospho-Vimentin (MBL International, Japan, D076-3s, 1:500), rat anti-BrdU (Abcam ab6326, 1:1,000), rabbit anti-phospho-histone H3 (ser10) (Santa Cruz sc-8656-R, 1:400), rabbit anti-N-cadherin (Abcam ab12221, 1:500), rabbit anti-GFAP (Dako z0334, 1:1,000), goat anti-Numb (Abcam ab4147, 1:400), rat anti-ITGB1 (Millipore MAB1997, 1:500), rabbit anti-pERK1/2 (Cell Signaling Technology, Danvers, MA, #4370, 1:500), rabbit anti-laminin (Sigma L9393, 1:400), rabbit anti-HOPX (Sigma HPA030180, 1:1000). Secondary antibodies were: AlexaFluor 488 (1: 1,000), 546 (1: 1,000), 594 (1: 500), or 647 (1: 1,000) -conjugated donkey anti-goat, -rabbit, -rat, -mouse IgG (Invitrogen), or -chicken (Sigma). All images were acquired on a Nikon A1R laser confocal microscope except that the labeled brain slices from time-lapse imaging were imaged on the Olympus FV10i-O with a 10x (zoom x2) air objective lens.

## In situ hybridization

Fresh mice brains were mounted in O.C.T. compound (Tissue-Tek) and frozen at −80°C to be sectioned coronally (30 μm) with a cryostat (Leica, Germany, CM1950). Cryosections were collected on superfrost plus microscope slides (Fisher Scientific, Pittsburgh, PA). To generate template cDNA for RNA probe synthesis, mouse total RNA was extracted from cortical samples of E15.5 fetus by standard Trizol (Life Technologies, Holliston, MA) method and reverse transcribed with QuantScript RT kit using oligo (dT)$_{15}$ primers (Tiangen). *Cdh2* gene was amplified using primers: 5'-CTGCCATGAC TTTCTACGG-3' (forward), 5'-GGTTGATGGTCCAGTTTC-3' (reverse). *TBC1D3* was amplified using primers: 5'-ATGGACGTGGTAGAGGTCGC-3' (forward), 5'-CTAGAAGCCTGGAGGGAACTG-3' (reverse). PCR products of predicted band size were gel extracted and ligated into the pGEMT Vector System (Promega, Madison, WI). Ligation products were transfected into DH5α competent E.coli (Tiangen) and confirmed by sequencing. Digoxigenin labeled RNA probes for in situ hybridization were generated by amplifying target DNA fragments from pGEMT vector using T7 or SP6 RNA Polymerase (Promega) in the presence of DIG RNA Labeling Mix (Roche, Switzerland). Synthesized antisense or sense RNA probes (1 ng/μl) were applied in hybridization. The detailed procedure for in situ hybridization was performed as described previously (*Wallace and Raff, 1999*). Images were collected with Nikon microscope ECLIPSE E600FN with an Optronics MicroFire digital camera.

## Mouse brain slice culture and time-lapse imaging

The detailed procedure for brain slice culture and time-lapse imaging was performed mainly as reported previously (*Wang et al., 2011*) with some modifications. E13.5 fetal mice brain cortices were electroporated with 0.6 μg/μl pCAGGS-TBC1D3-IRES-EGFP plasmids, or 1 μg/μl pCS2-Myc-TBC1D3 plus 0.5 μg/μl pCAG-H2BGFP-2A-tdTomato, with pCAGGS-IRES-EGFP or pCS2-MT as respective control. And at 24 hr post-electroporation, brain tissues were dissected out into ice-cold artificial cerebrospinal fluid (ACSF) containing 125 mM NaCl, 5 mM KCl, 1.25 mM NaH$_2$PO$_4$, 1 mM MgSO$_4$, 2 mM CaCl$_2$, 25 mM NaHCO$_3$ and 20 mM D-(+)-glucose (all from Sigma); pH 7.4, 310 mOsm1$^{-1}$. Brains were embedded into 3% low melting temperature agarose in ACSF and sectioned at 300 μm thickness using a Leica VT1200S vibratome. Then brain slices were transferred and collected temporarily in ice-cold ACSF pre-oxygenated with 95% O$_2$, 5% CO$_2$. Rostral brain slices containing EGFP-positive cells were selected and transferred onto a slice culture insert (Millicell, Millipore) in a glass-bottom Petri dish (Eppendorf, Germany) with pre-warmed (37°C) culture medium containing (v/v) 66% Eagle's basal medium, 25% Hanks balanced salt solution (without calcium and magnesium), 5% FBS, 1% N$_2$ supplement, 1% penicillin/streptomycin, 2 mM L-glutamine (all from Gibco) and 0.66% (w/v) D-(+)-glucose (Sigma). Brain slices were maintained in a humidified incubator at 37°C with constant 5% CO$_2$ supply for 2–3 hr before time-lapse imaging. All the time-lapse images in this study were collected in 20 min intervals and 14 hr duration by using Olympus laser confocal microscope FV10i-W with a Built-in incubator (37°C) streamed with 5% CO$_2$, 95% O$_2$ and an 10x (zoom x2) air objective lens. For identifying the types of daughter cells divided from oRG-like cells, brain slices, after time-lapse imaging, were fixed immediately in cold 4% PFA overnight and then stained using the method mentioned above.

## Culture and electroporation of embryonic human brain slices

Fresh embryonic brain samples obtained from voluntary abortions were treated mainly following the procedures described in a previous study (*Pollen et al., 2015*). Briefly, samples were transferred and dissected in filtered ACSF containing antibiotic antimycotic (Gibco) equilibrated with 5% CO$_2$, 95% O$_2$, embedded in 4% low melting point agarose (Invitrogen) and 300 μm coronal sections were prepared in the presence of ACSF using vibrating microtome. Brain slices were transferred into slice culture inserts (Millicell, Millipore) in 6-well culture plates (Corning, Corning, NY) with culture media containing 66% Eagle's basal medium, 25% Hanks balanced salt solution, 5% fetal bovine serum, 1% N-2 supplement, 1% antimycotic, and 1% glutaMAX supplement (all from Gibco), and equilibrated at 37°C in 95% O$_2$, 5% CO$_2$ for 2–3 hr. Then the brain slices were subjected to electroporation with 1–3 μg indicated plasmids using a pair of home-made electrode (five 50-ms 40 mV pulses with a 950-ms interval). After electroporation, slices were cultured in fresh medium in a 37°C incubator at 5% CO$_2$, 95% O$_2$ for three days, followed by fixation in 4% PFA overnight, three washes with PBS, antigen retrieval (Beyotime, China) for 4 hr at RT, permeablization with 2% Triton-X 100 in PBS at

4°C overnight, and finally immunostaining with indicated antibodies, which were diluted in blocking buffer containing 10% donkey serum, 0.5% Triton-X 100 and 0.2% gelatin in PBS.

## Fluorescence-activated cell sorting and quantitative real-time PCR

Mice brains (E15.5) were dissected out at 48 hr post-electroporation with pE/nestin-TBC1D3 or vehicle control, mixed with pCAG-YFP in a ratio of 3:1, and placed in cold ACSF pre-oxygenated with 95% $O_2$, 5% $CO_2$. The YFP-positve cortical region was removed using an inverted fluorescence microscope (Olympus, Japan, CKX41) and digested in 0.025% trypsin (Sigma) in ACSF for 20 min at 37°C and then centrifuged for 5 min at 1200 rpm. After removing the trypsin supernatant, tissue was re-suspended in 1 ml of ACSF containing 3% FBS (Hyclone, Logan, UT) and manually triturated by pipetting up and down approximately nine times. The suspension was passed through a 40-µm nylon cell strainer (BD Falcon, San Jose, CA) to obtain single-cell suspension and stored on ice before sorting. The YFP-positive cells were sorted into RNase-free tubers (Axygen, Corning, NY) on ice by using a MoFlo XDP flow cytometry (Beckman Coulter, Brea, CA). The sorted cells ($5 \times 10^4$–$2 \times 10^5$) were centrifuged for 10 min at 300 g and the supernatant was removed. Total RNA from sorted YFP-positive cells ($5 \times 10^4$–$2 \times 10^5$) was extracted immediately by using an RNeasy Micro Kit (Qiagen, Germany) and reverse transcribed with QuantScript RT kit using oligo(dT)$_{15}$ primers (Tiangen). Real-time PCR was performed by using the Agilent Mx3000P qPCR system with the SYBR Premix Ex Taq II (Takara). Quantification was performed by the delta cycle time method, with mouse GAPDH used for normalization. The specific primers are:

GAPDH, 5'-AGAGTGTTTCCTCGTCCCG-3' (forward), 5'-CCGTTGAATTTGCCGTGA-3' (reverse); Cdh2, 5'-CCCCAAGTCCAACATTTC-3' (forward), 5'-CGCCGTTTCATCCATACC-3' (reverse); Numb, 5'-TAGAGCGTAAACAGAAGCG-3' (forward), 5'-CACTGATGGACCAACAACT-3' (reverse); Hes1, 5'-TGACGGCCAATTTGCCTTTC-3' (forward), 5'-TTCCGCCACGGTCTCCACA-3' (reverse); Hes5, 5'-GCACCAGCCCAACTCCAA-3' (forward), 5'-TCAGGAACTGTACCGCCTCC-3' (reverse); Trnp1, 5'-CCCAGGAAGGGACGGCAGAA-3' (forward), 5'-CCTCGGGTAAGGGCGGTGA-3' (reverse).

## BrdU/EdU double labeling

Sequential 5-bromo-2'-deoxyuridine/5-ethynyl-2'-deoxyuridine (BrdU/EdU) double labeling was performed mainly as previously described with some modifications (Insolera et al., 2014). In utero electroporation was conducted at E13.5 according to the method described above. The pregnant female mouse was injected intraperitoneally (i.p.) with 50 mg/kg BrdU (Sigma) at E14.5 and 100 µg EdU (Life Technologies) at E15.5. After 2 hr, fetal mice brains were dissected and fixed immediately in cold 4% PFA in PBS at 4°C overnight, followed by dehydration in 20% sucrose in PBS at 4°C. For transgenic mice, BrdU and EdU were sequentially injected at E13.5 and E16.5, respectively. Frozen sections (50 µm) were collected in PBS as described above. BrdU staining was performed using the manufacturer's protocol (Abcam). In brief, brain sections were successively treated with 1 M HCl for 10 min on ice, 2 M HCl for 10 min at RT and another 20 min at 37°C to denature the DNA. Immediately after the acid incubations, sections were transferred into 0.1 M borate buffer, pH 9.0, for 10 min of neutralization at RT. After washes with PBS, brain sections were treated with 0.3% Triton X-100 in PBS for 30 min followed by incubation in 10% donkey serum in PBS for 1 hr at RT, followed by incubation with primary antibodies (mouse anti-BrdU, Life Technologies B35128, 1:200; chicken anti-GFP, Aves Lab #GFP-1020, 1:500) at 4°C overnight, washes and incubation with the appropriate secondary antibodies for 1–2 hr at RT. EdU staining was performed using a Click-iT Plus EdU Imaging Kit (Life Technologies C10640) immediately after the incubation of secondary antibodies.

## Transgenic mice

The detailed procedure for producing TBC1D3-transgenic mice was performed mainly as described previously (Behringer et al., 2013). Briefly, to obtain nestin-TBC1D3 fragments, pE/nestin-TBC1D3 plasmids were digested with DraIII and MfeI (New England BioLabs, Ipswitch, MA). The target fragments were purified directly from the gel using a Whatman (Pittsburgh, PA) S&S ELUTRAP Electro-Separation System and diluted into 3 ng/µl in injection buffer for microinjection. Then the nestin-TBC1D3 fragments were injected into the pronucleus of C57BL/6 mouse zygotes and the injected embryos were implanted into the oviducts of day 1 pseudopregnant foster females (ICR). The TBC1D3-transgenic mice were identified by PCR using tail genomic DNA with specific primers: 5'-

CCCACAACTCCGATTACTCAA -3' (forward, P1), 5' -CGCCTGTTCGCCTTCTAC -3' (reverse, P2). The analysis and maintenance for TBC1D3-transgenic mouse was performed in the background of C57BL/6.

### DiI labeling

Mouse brains were dissected out in cold PBS and fixed in cold 4% PFA in PBS at 4°C overnight, followed by removal of the meninges that covers the cortical surface. The DiI crystals (Life Technologies) were dissolved in 100% ethanol to a final concentration of 1 mg/ml in 1 ml final volume. Then each brain sample was transferred to 1 ml fresh 4% PFA in PBS added with 30 µl of 1 mg/ml DiI solution and incubated at 37°C for another 24 hr. After DiI labeling, brains were washed with PBS, sectioned on the vibratome (Leica VT1200S) into 100 µm slices, and stained with DAPI before mounting.

### Image processing, quantitative analysis, and presentation

All images were processed by ImageJ or Fiji software. For each experiment with mice or cells, at least 3 biological replicates were performed. Biological replicates are defined as independent experiments in cells or the same experiment with different mouse embryos. For all experiments, statistical tests were performed using SigmaStat software: one way ANOVA followed by *Student-Newman-Keuls* test for multiple comparisons, or one way ANOVA on Ranks followed with Dunn's test for multiple comparisons for raw data without passing normality test or equal variance test, except for *Figure 2—figure supplement 2B and D* where two-way ANOVA followed with *Student-Newman-Keuls* test for multiple comparisons was used. Results presented as dot plots were accompanied by the mean value (red line) and a box plot in the background depicting confident interval. The sample size (n), the mean, the standard error of the mean (SEM), and the p-value are presented in the figure legends. The statistical significance was indicated by: *p<0.05; **p<0.01; ***p<0.001; ns, no significant difference.

## Acknowledgements

This study was partially supported by grants from the National Key Basic Research Program of China (2014CB910203), National Natural Science Foundation of China (31330032, 31490591, 31321091, and 61327902), and the Strategic Priority Research Program of the Chinese Academy of Sciences (XDB02040003). We are grateful to Dr. MM Poo for critical reading of the manuscript. We thank L Li for the assistance in the generation of transgenic mice, Dr. Q Hu of ION Imaging Facility with microscope analysis, Dr. X Zhang for expert assistance with in situ hybridization, Dr. X Yu for providing the N-cadherin constructs, and Dr. H Okano for providing the pE/nestin-EGFP construct.

## Additional information

### Funding

| Funder | Grant reference number | Author |
|---|---|---|
| Ministry of Science and Technology of the People's Republic of China | 2014CB910203 | Zhen-Ge Luo |
| National Natural Science Foundation of China | 31330032 | Zhen-Ge Luo |
| National Natural Science Foundation of China | 31490591 | Zhen-Ge Luo |
| National Natural Science Foundation of China | 31321091 | Zhen-Ge Luo |
| National Natural Science Foundation of China | 61327902 | Zhen-Ge Luo |
| Chinese Academy of Sciences | XDB02040003 | Zhen-Ge Luo |

The funders had no role in study design, data collection and interpretation, or the decision to submit the work for publication.

## Author contributions

X-CJ, Q-QH, Final approval of the version to be published, Conception and design, Acquisition of data, Analysis and interpretation of data, Drafting or revising the article; A-LS, Final approval of the version to be published, Acquisition of data, Analysis and interpretation of data, Contributed unpublished essential data or reagents; K-YW, Final approval of the version to be published, Acquisition of data, Analysis and interpretation of data; YZ, YJ, TW, ZY, Final approval of the version to be published, Analysis and interpretation of data, Contributed unpublished essential data or reagents; XW, Final approval of the version to be published, Conception and design, Acquisition of data, Analysis and interpretation of data; Z-GL, Final approval of the version to be published, Conception and design, Analysis and interpretation of data, Drafting or revising the article

## Author ORCIDs

Zhen-Ge Luo, http://orcid.org/0000-0001-5037-0542

## Ethics

Human subjects: Human fetal brain tissue samples were collected at autopsy within 3 hr of spontaneous abortion with the informed consent of the patients following protocols and institutional ethic guidelines approved by ethics committee of Shanghai Institutes for Biological Sciences, Chinese Academy of Sciences (Approval identifier number: ER-SIBS-221506). Brain tissues were stored in ice-cold Leibowitz-15 medium and transported to the laboratory for further examination and processing.

Animal experimentation: This study was performed in strict accordance with the recommendations in the Guide for the Care and Use of Laboratory Animals of the National Institutes of Health (8th Edition, 2010). All of the animals were handled according to approved institutional animal care and use committee (IACUC) protocols (#NA-008-2016) of the Institute of Neuroscience, Chinese Academy of Sciences. All surgery was performed under anesthesia with pentobarbital sodium, and every effort was made to minimize suffering.

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
