## [Decision Letter]

Thank you for resubmitting your work entitled "The Hominoid-specific gene TBC1D3 promotes generation of basal neural progenitors and induces cortical folding in mice" for further consideration at *eLife*. Your revised article has been favorably evaluated by a Senior Editor, a Reviewing Editor, and three reviewers, one of whom, Franck Polleux, has agreed to reveal his identity.

The manuscript has been improved but there are some remaining issues that need to be addressed before acceptance, as outlined below:

Reviewer 2 still has concerns, shared by the other reviewers upon consultation, about specificity of finding, that TBC1D3 could be regulating attachment to ventricular surface rather than fate, resulting in aberrant proliferation in basal location. Please see the comment from reviewer 2 below. The HOPX marker has been identified by several groups and immunostaining works well and is specific in human tissue, so you might consider evaluating this marker. One reviewer pointed out an example of a recent publication on CYFIP1 manipulation, where a similar phenotype of enhanced proliferation of progenitors away far ventricles without an attachment to the ventricle was observed, and where they reached a conclusion quite distinct from the one that you present (i.e. they did not conclude CYFIP1 deletion converts vRG to oRG).

*eLife* would like you to address this point in the discussion, or if you have new experimental data to address this point, consider to include it into the manuscript.

Reviewer #1:

The revised version of this manuscript is satisfactory; the reviewers have clearly improved the figures and the text by addressing most of the comments made initially. I would like to approve this manuscript for publication in *eLife*.

Reviewer #2:

In this revised manuscript, Ju et al. responded to previous critics with new experiments and many of the previous concerns are addressed with new data. In addition, the authors provided new data on loss-of-function analysis in human fetal tissue and signaling mechanism in mouse model. They also removed the behavioral analysis result. The key conclusion the authors made is that TBC1D3 is required for oRG generation in regulating cortical expansion and folding. However, the conclusion on oRG is mainly made with morphological features (basal localization, lack of attachment to the ventricular surface), not by definitive molecular signatures/markers. It is still possible that TBC1D3 simply regulates attachment to ventricular surface, lack of which leads to aberrant proliferation in basal location. This does not necessarily mean conversion of ventricular progenitors to oRG cells. Recently, there are multiple markers identified for oRG cells. While these markers may not work for mouse, they should be tested in their human fetal tissue studies.

Reviewer #3:

The authors have addressed most of my questions. I support the publication.

[Editors note: a previous version of this study was rejected after peer review, but the authors submitted for reconsideration. The first decision letter after peer review is shown below.]

Thank you for submitting your work entitled "Expression of a hominoid-specific gene in mice induces cortical folding and enhances motor learning" for consideration by *eLife*. Your article has been reviewed by three peer reviewers, and the evaluation has been overseen by a Reviewing Editor and a Senior Editor. Our decision has been reached after consultation between the reviewers. Based on these discussions and the individual reviews below, the experiments required to address the major points raised by the reviewers would exceed the time limit imposed by *eLife* guidelines, so we regret to inform you that your work will not be considered further for publication in *eLife*.

Increased population of oRG in human OSVZ has been recently suggested as a potential mechanism for cortical expansion that differentiates brain structures between human and rodent. In this study, Ju et al. investigated the function of TBC1D3, a hominoid-specific gene, in regulating the expansion and folding of the mammalian neocortex. TBC1D3 has previously been shown to be a hominoid-specific oncoprotein that promotes cell proliferation (Pei et al., 2002), activates RAS and enhances EGF/EGFR and insulin signaling (Wainszelbaum et al., 2008; Wainszelbaum et al., 2012).

Here, the authors expressed TBC1D3 in the developing mouse cortex using in utero electroporation and transgenic, and found that (over)expression of TBC1D3 led to a disruption of the junction at the ventricular zone surface, a delamination of radial glial progenitors, increase in self-renewable outer radial glia (oRGs) and intermediate progenitors, and an effect on motor strip cortical folding. Moreover, the authors showed that the TBC1D3 transgenic mice exhibit an improvement in a certain aspect of motor learning compared with the control. The authors also suggested a potential link between TBC1D3 and TRNP1, a DNA-binding protein that has previously been shown to regulate cortical progenitor behavior and folding.

The impression of the reviewers and the board is that this manuscript addresses an interesting and timely topic in the field and the authors performed analyses to support the main conclusions. The finding that TBC1D3 overexpression is sufficient to lead to some gyrification of the rodent cortex is interesting, but questions from reviewers about the underlying mechanisms of gyrification, and the relevance of the motor learning enhancement remain. The issues raised by the reviewers would, in our opinion, require more time than the 2 months maximum allowed at *eLife* for to receive revisions, and thus *eLife* has decided to reject the manuscript but is willing to consider a revision to be returned to the reviewers if these issues can be addressed sufficiently. The number of criticisms and the time that might be required to address these criticisms might be more than you are willing to accept to see the work published in *eLife*, and thus you may decide to seek submission to an alternative journal.

Major points:

The authors should carefully characterize the ventricular zone organization (e.g. junction integrity), radial glial fiber scaffold, and pial basement membrane integrity. Some of the phenotypes presented may be due to disruption of the integrity of these structures, which can lead to many secondary effects, in which case the gyrification phenotype may be an artifact of anatomic disruption. Figure 3: The authors indicated a defect at the ventricular zone surface in the TBC1D3 section do the authors find a spatial correlation between the presence of "oRG-like cells" (and/or cell clumps) and the ventricular zone surface defect? Since TBC1D3 is most highly expressed at the surface of VZ, TBC1D3 might not directly transform vRG to oRG.

It appears that TBC1D3 expression exerts a strong non-cell autonomous effect (Figure 2,Figure 3 and supplemental figures) reflected in the electroporation experiments. For all these analyses, the authors should quantify cell autonomous (EYFP^+^) and non-cell autonomous (EYFP^-^) effects, e.g. in Figure 2, BrdU and PHH3 increase/decrease in EYFP^+^ or EYFP^-^ cells? Could the non-cell autonomous effect arise from the disruption of junction and disorganization of the ventricular zone? In principle, the disruption of junction could result in a detachment of radial glia (to be oRG-like; but not really oRG produced by RG division; Figure 3), clumps of cells, non-cell autonomous effects, and neuronal migration and organization defects. However, this line of processes may not really be related to neocortical expansion and folding.

The subcellular localization of TBC1D3 is mainly cytosolic in the developing human brain, as shown in Figure 1. The cytosolic role of overexpression TBC1D3 is further supported a lack of proliferation-increasing effect without cytosolic retention signal (?286-353, Figure 2). The authors suggested a transcriptional suppression of N-cadherin and Trnp1 by TBC1D3, but the altered expressions on those genes may be just indirect effects of cytosolic role of TBC1D3. Overexpressed TBC1D3 in human carcinoma cell lines is exclusively cytosol, via direct protein-protein interaction with microtubule (He et al. PLoS One 2014). Do the authors think that TBC1D3 directly regulates gene transcription? What exactly is ectopic TBC1D3 doing in vRG and oRG. How might the cytoplasmic retention influence its function? How may the phenotype that the authors observed in the mouse cortex be related to the known functions of TBC1D3, e.g. activating RAS and enhancing EGF and Insulin signaling?

How were quantification described Figure 3 obtained? How were the numbers of total RGs and oRG-like cells defined in 3F? (e.g. There are 3 DiI-labeled cells in the right panel (TBC1D3) of 3D. Was the one in the upper layer having both basal and apical process counted as an oRG or a soma translocating neuron?). In 3G, does # of events described indicate total dividing cells in a certain area of the imaged slice? The method of quantification should be clearly described.

In Figure 3, the authors perform IUE electroporation of TBC1D3 followed by DiI pial surface labeling to back-label potential outer Radial Glial cells (with an basal endfeet touching the pia but lacking an apical attachment in the ventricular zone). How can the authors observe such a significant effect on oRG-like cell density following DiI labeling since the IUE electroporation is usually sparse. The only potential explanation is that TBC1D3 has significant cell non-autonomous effects on non-electroporated cells. The authors should at least discuss this or perform DiI back-labeling on embryos from the transgenic mouse line where presumably all the neural progenitors in the dorsal telencephalon express TBC1D3.

The behavioral analysis is inadequate to conclude that motor learning is enhanced. Because results of rotarod test in Figure 5 is the only evidence that the gene or cortical folding enhances learning, it would be better to provide further information or supportive evidence. Figure 8 is not convincing and the phenotypes shown do not relate well to the rest of the paper. It's not clear why increasing the number of neurons in superficial layers of the cortex would be responsible for the rotarod phenotype shown in Figure 8. Why did authors use average foot position from the top instead of latency to fall, which is widely used and more distinct? This type of motor learning involving balance on the rotator is generally thought to require changes in cerebellar circuits, which was not explore here. Behavioral tests usually more dependent on hippocampal and cortical circuits include novel object recognition or some sensory discrimination/associative learning tests. The authors should either perform a broader range of behavioral tests in order to determine if there is a general improvement in 'cognitive' or 'motor' performance in these transgenic mouse line(s) and more carefully interpret the results in the context of the motor learning field.

---

## [Author Response]

*Reviewer 2 still has concerns, shared by the other reviewers upon consultation, about specificity of finding, that TBC1D3 could be regulating attachment to ventricular surface rather than fate, resulting in aberrant proliferation in basal location. Please see the comment from Reviewer 2 below. The HOPX marker has been identified by several groups and immunostaining works well and is specific in human tissue, so you might consider evaluating this marker. One reviewer pointed out an example of a recent publication on CYFIP1 manipulation, where a similar phenotype of enhanced proliferation of progenitors away far ventricles without an attachment to the ventricle was observed, and where they reached a conclusion quite distinct from the one that you present (i.e. they did not conclude CYFIP1 deletion converts vRG to oRG).*

*eLife would like you to address this point in the discussion, or if you have new experimental data to address this point, consider to include it into the manuscript.*

These concerns are reasonable. We have addressed the remaining issues by adding new experimental data and through discussion.

We performed immunostaining with HOPX antibody for stored brain slices from E14.5 WT and TG mice. Surprisingly, we found that TG mice exhibited many HOPX^+^ cells in extra-VZ basal regions, whereas these cells were barely observed in WT control mice (see revised Figure 6). This result strongly supports our conclusion that TBC1D3 expression promotes the generation of oRG cells in mice.

We believe that simple ectopic localization of RG cells outside of the ventricular zone may not be sufficient to cause cortical expansion or folding as shown in the mentioned CYFIP1 study. Appearance of oRG-like features, including the contact with pia surface by the basal process, multiple rounds of cell division, expression of specific markers, as well as clustered regional distribution of BPs observed in our study, may play prominent roles in cortical folding. This point has been discussed in the revised manuscript (see the end of third paragraph of Discussion section.

*Reviewer #2:*

*In this revised manuscript, Ju et al. responded to previous critics with new experiments and many of the previous concerns are addressed with new data. In addition, the authors provided new data on loss-of-function analysis in human fetal tissue and signaling mechanism in mouse model. They also removed the behavioral analysis result. The key conclusion the authors made is that TBC1D3 is required for oRG generation in regulating cortical expansion and folding. However, the conclusion on oRG is mainly made with morphological features (basal localization, lack of attachment to the ventricular surface), not by definitive molecular signatures/markers. It is still possible that TBC1D3 simply regulates attachment to ventricular surface, lack of which leads to aberrant proliferation in basal location. This does not necessarily mean conversion of ventricular progenitors to oRG cells. Recently, there are multiple markers identified for oRG cells. While these markers may not work for mouse, they should be tested in their human fetal tissue studies.*

We thank the reviewer for the constructive suggestion. We performed immunostaining with HOPX antibody for stored brain slices from E14.5 WT and TG mice. Surprisingly, we found that TG mice exhibited many HOPX^+^ cells in extra-VZ basal regions, whereas these cells were barely detectable in WT control mice (see revised Figure 6). This result strongly supports our conclusion that TBC1D3 expression promotes the generation of oRG cells in mice. Given that vast majority of *Sox2*^+^ cells are positive for recently identified oRG markers (Pollen AA et al. Cell 2015; Thomsen ER et al. Nat. Methods 2016), our results that TBC1D3 knock-down decreased the generation of *Sox2*^+^ in SVZ regions supports our conclusion that TBC1D3 plays a critical role in oRG generation during human cortex development. Related text has been modified accordingly.

[Editors note: the author responses to the first round of peer review follow.]

*Major points:*

*The authors should carefully characterize the ventricular zone organization (e.g. junction integrity), radial glial fiber scaffold, and pial basement membrane integrity. Some of the phenotypes presented may be due to disruption of the integrity of these structures, which can lead to many secondary effects, in which case the gyrification phenotype may be an artifact of anatomic disruption. Figure 3: The authors indicated a defect at the ventricular zone surface in the TBC1D3 section do the authors find a spatial correlation between the presence of "oRG-like cells" (and/or cell clumps) and the ventricular zone surface defect? Since TBC1D3 is most highly expressed at the surface of VZ, TBC1D3 might not directly transform vRG to oRG.*

These are reasonable concerns. We performed several additional experiments to address these questions. First, we found that expression of TBC1D3 caused dislocalization of Numb and integrin ?1 (ITGB1) which, were originally polarized distributed in endfeet of vRG cells (Campos et al., 2004; Katayama et al., 2011; Rasin et al., 2007), without disrupting the junction integrity of the ventricular zone as revealed by actin filaments (F-actin) (see revised Figure 1—figure supplement 2). Second, DiI tracing shows that radial glial fiber scaffold is intact in TBC1D3 transgenic mice (revised Figure 6). Third, TBC1D3 electroporation has no effect on the pial basement membrane integrity as revealed by staining with laminin (see revised Figure 5—figure supplement 1). Thus, the observed cortical folding was not due to disruption of the integrity of these structures. Indeed, delamination itself is not sufficient for the generation of real oRG-like cells. As shown in revised Figure 2—figure supplement 3, detachment of vRG cells induced by blocking N-cadherin-mediated adhesion does not promote generation of basal progenitors. The concerted effects on cell adhesion and stemness pathways, including N-cadherin, Trnp1, and Ras/ERK signaling (see below), may render TBC1D3's function in promoting oRG generation.

To further determine the physiological role of TBC1D3, we performed loss-of-function study in cultured human brain slices, and found that TBC1D3 is required for the generation of oRG cells or basal progenitors (see revised Figure 4).

*It appears that TBC1D3 expression exerts a strong non-cell autonomous effect (Figure 2,Figure 3 and supplemental figures) reflected in the electroporation experiments. For all these analyses, the authors should quantify cell autonomous (EYFP^+^) and non-cell autonomous (EYFP^-^) effects, e.g. in Figure 2, BrdU and PHH3 increase/decrease in EYFP^+^ or EYFP^-^ cells? Could the non-cell autonomous effect arise from the disruption of junction and disorganization of the ventricular zone? In principle, the disruption of junction could result in a detachment of radial glia (to be oRG-like; but not really oRG produced by RG division; Figure 3), clumps of cells, non-cell autonomous effects, and neuronal migration and organization defects. However, this line of processes may not really be related to neocortical expansion and folding.*

The seemingly non-cell autonomous effect of TBC1D3 expression on proliferation of BPs was most likely caused by different dosage of pE/nestin-TBC1D3 and pCAG-YFP (3:1) used in electroporation, because electroporation with pCAGGS-TBC1D3-IRES-EGFP, a vector co-expressing both TBC1D3 and EGFP, caused increase in numbers of basal BrdU^+^ or PH3^+^ cells in EGFP^+^ but not in EGFP^-^ cells (see revised Figure 2—figure supplement 2). The notable slight decrease of EGFP^-^ apical neural progenitors might be due to the disruption of proliferation niche in VZ regions (see revised Figure 2—figure supplement 2, left). Indeed, detachment of vRG cells is not sufficient for the generation of real oRG-like cells. As shown in revised Figure 2—figure supplement 3, detachment of vRG cells induced by blocking N-cadherin-mediated adhesion does not promote generation of basal progenitors.

*The subcellular localization of TBC1D3 is mainly cytosolic in the developing human brain, as shown in Figure 1. The cytosolic role of overexpression TBC1D3 is further supported a lack of proliferation-increasing effect without cytosolic retention signal (?286-353, Figure 2). The authors suggested a transcriptional suppression of N-cadherin and Trnp1 by TBC1D3, but the altered expressions on those genes may be just indirect effects of cytosolic role of TBC1D3. Overexpressed TBC1D3 in human carcinoma cell lines is exclusively cytosol, via direct protein-protein interaction with microtubule (He et al. PLoS One 2014). Do the authors think that TBC1D3 directly regulates gene transcription? What exactly is ectopic TBC1D3 doing in vRG and oRG. How might the cytoplasmic retention influence its function? How may the phenotype that the authors observed in the mouse cortex be related to the known functions of TBC1D3, e.g. activating RAS and enhancing EGF and Insulin signaling?*

We don't think that TBC1D3 directly regulates gene transcription. We have adjusted this point in the revised manuscript. First, we performed additional experiments to determine how does TBC1D3 regulate N-cadherin and Trnp1 transcripts and found that expression of TBC1D3 indeed caused destabilization of N-cadherin transcript (see revised Figure 1—figure supplement 1 and Figure 4). The decrease in the level of Trnp1 might also be indirect. We propose that the expression of TBC1D3 in the VZ leads to detachment of vRGs via downregulation of N-cadherin and/or Trnp1, and promotes proliferation of oRG-like cells through Ras-ERK signaling. Several newly added lines of evidence support this model:

First, co-expression of a dominant-negative form of Ras^S17N^ (Ras-DN) inhibited proliferation of TBC1D3-induced basal progenitors (see revised Figure 2—figure supplement 4); Second, phosphorylated active ERK (pERK) was up-regulated in TBC1D3 transgenic mice; Third, pERK was mostly present in Pax6^+^Tbr2^-^ oRG cells in the developing human brain (see revised Figure 7figure supplement 2). Thus, TBC1D3 promotes oRG generation and proliferation through concerted actions on multiple molecules.

*How were quantification described Figure 3 obtained? How were the numbers of total RGs and oRG-like cells defined in 3F? (e.g. There are 3 DiI-labeled cells in the right panel (TBC1D3) of 3D. Was the one in the upper layer having both basal and apical process counted as an oRG or a soma translocating neuron?). In 3G, does # of events described indicate total dividing cells in a certain area of the imaged slice? The method of quantification should be clearly described.*

This point has been clarified in the revised Figure 3. We quantified the percentage of oRG-like cells with a basal process attached to the pial surface and soma located in SVZ or IZ regions, among total RGs including vRGs with soma located in the VZ (see revised Figure 3 legend). The one on the upper layer having both apical and basal processes was not counted as oRG, because its soma was localized in the cortical plate but not in IZ/SVZ (revised Figure 3).

Quantification for the relative number of basal progenitors with oRG-like divisions, identified by time-lapse imaging in Figure 3, was based on the results from per unit of tangential length of the VZ electroporated (see revised Figure 3 legend).

*In Figure 3, the authors perform IUE electroporation of TBC1D3 followed by DiI pial surface labeling to back-label potential outer Radial Glial cells (with an basal endfeet touching the pia but lacking an apical attachment in the ventricular zone). How can the authors observe such a significant effect on oRG-like cell density following DiI labeling since the IUE electroporation is usually sparse. The only potential explanation is that TBC1D3 has significant cell non-autonomous effects on non-electroporated cells. The authors should at least discuss this or perform DiI back-labeling on embryos from the transgenic mouse line where presumably all the neural progenitors in the dorsal telencephalon express TBC1D3.*

As shown in revised Figure 2—figure supplement 2, TBC1D3 does not have non-autonomous effects on the generation and proliferation of basal progenitors. The observed significant effect on oRG-like cell density most likely attributed to regional overexpression of TBC1D3 in a large number of vRGs. As suggested by the reviewer, we performed DiI back-labeling on embryos from the TBC1D3 transgenic mice and again found a marked increase in oRG-like cells (see revised Figure 6).

*The behavioral analysis is inadequate to conclude that motor learning is enhanced. Because results of rotarod test in Figure 5 is the only evidence that the gene or cortical folding enhances learning, it would be better to provide further information or supportive evidence. Figure 8 is not convincing and the phenotypes shown do not relate well to the rest of the paper. It's not clear why increasing the number of neurons in superficial layers of the cortex would be responsible for the rotarod phenotype shown in Figure 8. Why did authors use average foot position from the top instead of latency to fall, which is widely used and more distinct? This type of motor learning involving balance on the rotator is generally thought to require changes in cerebellar circuits, which was not explore here. Behavioral tests usually more dependent on hippocampal and cortical circuits include novel object recognition or some sensory discrimination/associative learning tests. The authors should either perform a broader range of behavioral tests in order to determine if there is a general improvement in 'cognitive' or 'motor' performance in these transgenic mouse line(s) and more carefully interpret the results in the context of the motor learning field.*

We agree with reviewer 3 that "behavioral tests usually more dependent on hippocampal and cortical circuits include novel cognition or some sensory discrimination/associative learning tests" and it is difficult to relate the increased number of neurons in superficial layers of the cortex to the rotarod phenotype, which may also require changes in cerebellar circuits. All these analyses are out of the scope of this work, and thus we choose not including the behavior data in the revised manuscript.